# Transformers learn factored representations

**Adam Shai** [1]  **Loren Amdahl-Culleton** [1]  **Casper L. Christensen** [1]  **Henry R. Bigelow** [1]  **Fernando E. Rosas** [2] [3] [4]
**Alexander B. Boyd** [5]  **Eric A. Alt** [1]  **Kyle J. Ray** [1]  **Paul M. Riechers** [1]

## Abstract

Transformers pretrained via next-token prediction learn to factor their world into parts, representing these factors in orthogonal subspaces of the residual stream. We formalize two representational hypotheses: (1) a representation in the product space of all factors, whose dimension grows exponentially with the number of parts, or (2) a factored representation in orthogonal subspaces, whose dimension grows linearly. The factored representation is lossless when factors are conditionally independent, but sacrifices predictive fidelity otherwise, creating a tradeoff between dimensional efficiency and accuracy. We derive precise predictions about the geometric structure of activations for each, including the number of subspaces, their dimensionality, and the arrangement of context embeddings within them. We test between these hypotheses on transformers trained on synthetic processes with known latent structure. Models learn factored representations when factors are conditionally independent, and continue to favor them early in training even when noise or hidden dependencies undermine conditional independence, reflecting an inductive bias toward factoring at the cost of fidelity. This provides a principled explanation for why transformers decompose the world into parts, and suggests that interpretable low dimensional structure may persist even in models trained on complex data.

[1]Simplex, Astera Institute [2]Sussex AI and Sussex Centre for Consciousness Science, University of Sussex [3]Centre for Complexity Science and Center for Psychedelic Research, Department of Brain Sciences, Imperial College London [4]Center for Eudaimonia and Human Flourishing, University of Oxford [5]Beyond Institute for Theoretical Science (BITS). Correspondence to: Adam Shai <adamimos@gmail.com>, Paul Riechers <pm-riechers@gmail.com>.

*Proceedings of the $43^{rd}$ International Conference on Machine Learning*, Seoul, South Korea. PMLR 306, 2026. Copyright 2026 by the author(s).

## 1. Introduction

It's easy to take for granted that our world is not monolithic, but is made of parts. But for ourselves and AI models like transformers, we merely perceive the world via a stream of undifferentiated inputs. Humans take this stream of raw data, and decompose it into discrete objects—chairs, coffee mugs, other agents—in order to understand the world. Do transformers also learn to factor their world into parts? How can they from a mere stream of tokens, and why would they?

In this paper, we show that neural networks like transformers indeed learn and have an inductive bias for factored representations as a natural consequence of next-token pretraining. We rigorously test the nature of factorization in transformer representations by presenting a formal framework connecting training data structure to factored representations in neural networks. We use this framework to generate competing geometric hypotheses, and our experiments test which geometry transformers learn. Overall, this work brings three main contributions:

1. **A theoretical framework linking latent factorization in the data-generating process to activation geometry**. This provides a rigorous bridge between compositional structure in the data-generating process and properties of the activations of neural networks, including the existence of orthogonal subspaces, their dimensionality, and the geometry of context embeddings within them.

2. **Empirical confirmation that transformers learn factored structure.** When the data-generating process admits a decomposition into conditionally independent factors, transformers learn factored representations in orthogonal subspaces, achieving the predicted exponential dimension reduction even when model capacity would easily accommodate the full unfactored representation.

3. **Evidence for an inductive bias toward factored representations.** When the data-generating process is not factorizable, transformers nevertheless first learn a lossy factored representation before gradually expanding dimensionality to recover fidelity. The factored solution acts as a representational attractor: models

find it quickly, dwell there, and depart only under sustained pressure from the loss gradient. This reveals that transformers actively seek to factorize representations more generally.

Figure 1 provides an overview of the framework and main results.

## 2. From training data to activation geometry

In order to test if transformers have representations that correspond to parts in some way, we begin by establishing a theoretical framework which mathematically captures the relationship between the sequences of tokens that make up training data, and the expected internal representations in networks trained to do next-token predictions on that data. First, we establish what constitutes *latent structure* in training data, and from there define what that structure looks like in a factored and non-factored form. Our testable predictions then come from establishing what geometry is implied by the task of next-token prediction on those two generators.

We build directly on the **activation geometry** program introduced by Shai et al. (2024), which established a relationship between the latent structure of a training data generator in a hidden Markov model setting and the geometric structure of activations in a transformer's residual stream. That work tested a key distinction: the objects relevant to optimal prediction are not the generator's hidden states, but the observer's context-dependent belief in the form of probability distributions over those hidden states (Upper, 1997; Riechers & Crutchfield, 2018; Marzen & Crutchfield, 2017). Each context induces such a belief, and the collection of beliefs across all contexts forms a geometry where nearby points make similar predictions about the entire future token sequence.

If belief geometry emerges when training on a single latent generator, what geometry emerges when the generator is made of parts? We show that a generator built from parts admits two natural representations of its predictive vector: a joint one, living in the tensor product of the parts' spaces, and a factored one, living in their direct sum. Both are consistent with optimal prediction, but the two correspond to different, precisely specified geometries. The geometry a trained network exhibits in its internal activation space therefore adjudicates between them. The rest of this section makes both precise and derives what each predicts for transformer activations.

### 2.1. Latent data generators

What constitutes the structure of training data in the context of standard next-token prediction, and how might it

constrain what a network learns? Consider length-$L$ token sequences $x_{1:L} = (x_1, \ldots, x_L) \in \mathcal{X}^L$, sampled from a ground-truth joint distribution $Q(X_{1:L})$. These sequences will generally be non-Markovian, such that predicting the next-token optimally may require information in arbitrarily long contexts.

We conceptualize these sequences as being generated by a latent process that evolves over time and emits observed tokens. To make this precise, we use generalized hidden Markov models (GHMMs). Any distribution over sequences admits a GHMM representation, so this is fully general. We will use this to derive exact predictions about transformer representations in the following subsections. A GHMM is defined by the tuple:

$$\mathcal{M} = \left( \mathcal{X}, \mathcal{S}, \boldsymbol{\eta}^{(\varnothing)}, (T^{(x)})_{x \in \mathcal{X}} \right) \tag{1}$$

where $\mathcal{X}$ is the token alphabet, $\mathcal{S} = \mathbb{R}^d$ is the latent space, $\boldsymbol{\eta}^{(\varnothing)}$ is an initial state vector in $\mathbb{R}^d$, and $T^{(x)}$ is the operator describing latent dynamics for each token $x$. For GHMMs, the probability of any sequence can be calculated as

$$Q_{\mathcal{M}}(x_{1:\ell}) = \boldsymbol{\eta}^{(\varnothing)} T^{(x_1)} \cdots T^{(x_\ell)} \mathbf{1} , \tag{2}$$

where $\mathbf{1}$ is the vector that integrates probability in the latent space. GHMMs include hidden Markov models (HMMs) as a special case[1], but also include more general generators. For details, see Appendix A.1.

### 2.2. Geometry associated with latent generators

After observing a context $x_{1:L}$, what does the model need to know to predict optimally? Only the information the past reveals about the future (Upper, 1997). We call the vector encoding this information the *predictive vector*:

$$\boldsymbol{\eta}^{(x_{1:\ell})} := \frac{\boldsymbol{\eta}^{(\varnothing)} T^{(x_1)} \cdots T^{(x_\ell)}}{\boldsymbol{\eta}^{(\varnothing)} T^{(x_1)} \cdots T^{(x_\ell)} \mathbf{1}} . \tag{3}$$

Two contexts with identical predictive vectors make identical predictions about all future tokens. Contexts with *similar* predictive vectors make *similar* predictions, and prior work has shown this similarity structure is reflected geometrically in transformer activations (Shai et al., 2024).

The collection of predictive vectors across all possible contexts forms a geometric arrangement in the latent space, which is determined by the structure of the data-generating process (Figure 1a, bottom). For a $d$-dimensional latent space, this arrangement takes place in $d - 1$ dimensions (since predictive vectors are normalized). For more discussion and examples, see Appendices A.2 and B.

---

[1]In this case, there are $d$ hidden states, and substochastic transition matrices $T^{(x)} : \mathbb{R}^d \to \mathbb{R}^d$ whose matrix elements are simply token-dependent transition probabilities between latent states. This provides helpful intuition for many (but not all) purposes.

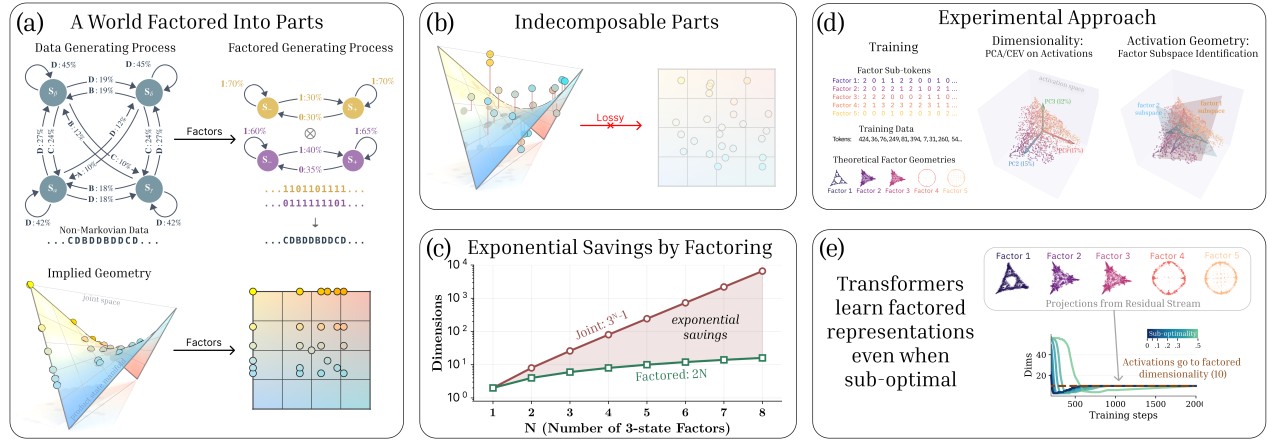

*Figure 1.* Transformers learn to factor the world into parts, representing each factor in orthogonal subspaces. **(a)** Consider a data-generating process with 4 hidden states (top-left). Although the joint process appears complex, it may admit a representation as a product of simpler independent processes running in parallel (top-right). Below, we illustrate how this factorization affects the geometry of representations: the full simplex over joint latent states (a tetrahedron for four joint states) decomposes into the product of simplices for each factor (two line segments). Even in cases where parts influence each other's dynamics, as long as conditioning on observed tokens renders the factors independent, the same geometric decomposition applies: uncertainty over each factor can be tracked separately in orthogonal subspaces. **(b)** When factors remain correlated even after conditioning on observations, the joint state cannot be written as a product state—it lies off the product-state manifold. Projecting onto the factored representation (red arrow) is lossy: some predictive information is sacrificed for the dimensional savings. **(c)** This decomposition yields dramatic representational savings. For $N$ three-state factors, the joint representation requires $3^N - 1$ dimensions (red), while the factored representation requires only $2N$ dimensions (green)—an exponential reduction. **(d)** To determine if transformers learn factored representations, we construct training data that is latently composed of subtokens generated from 5 factors. Each of these factors has a geometric prediction from our theory. Our analysis methods consist of quantifying the dimensionality of the activations using PCA and explained variance, as well as determining the geometric arrangement of factor subspaces. **(e)** Our main results are that transformers learn factored representations, placing these representations in orthogonal subspaces of the residual stream. Crucially, transformers do this even when it hurts prediction.

## 2.3. Conditionally independent data-generators

Consider a data-generating process that has *parts*, composed of multiple processes running in parallel. An example is shown in Figure 1a. Each part has its own latent space, and each contributes to the observed output. **Crucially, these parts are not visible at the token level.**[2] The factorization into parts exists purely in the latent structure of the generator. Discovering it requires inferring, from the correlational structure of token sequences alone, that the world decomposes into parts.

We call such generators *conditionally independent*: given the observed token, each factor's latent dynamics evolve independently. The factors may influence each other's outputs, but once you condition on what was actually emitted, they decouple. This can be formalized in terms of GHMMs:

**Definition 2.1.** *A GHMM generator is **conditionally independent** if each token-labeled linear operator can be expressed as the tensor product of multiple linear operator*

---

[2]Through training, the model sees only the token, and nothing signals that, e.g., 'token 347' is secretly 'subtoken 2 from process 1, subtoken 0 from process 2, etc'.

*factors:*

$$T^{(x)} = \bigotimes_{n=1}^{N} T_n^{(x)} \ . \tag{4}$$

Such a multipartite generator has latent space $\mathcal{S} = \bigotimes_{n=1}^{N} \mathcal{S}_n$, the tensor product of $N$ constituent latent spaces. The dimension of this joint latent space grows exponentially with the number of components. $N$ factors with $m$-dimensional latent space yield a joint space of dimension $m^N$.

Conditionally independent generators realize a wide variety of models of interest. In our experiments, we consider $N$ processes that emit unobserved subtokens $x^{(1)}, \ldots, x^{(N)}$, each with their own generator. The observed vocabulary $\mathcal{X}$ has a unique token for each collection of subtokens; nothing in the training data explicitly signals the underlying factored structure.

## 2.4. Joint vs. factored representations

We now arrive at our central question: what representational geometry should we expect to find in a network trained on data from a world with parts? Two possibilities are mathematically natural:

**The joint representation** tracks predictive vectors in the full tensor-product space (Figure 1a, bottom left). For $N$ factors with latent dimensions $d_1, \ldots, d_N$, this requires $(\prod_n d_n) - 1$ dimensions, but is always lossless.

**The factored representation** tracks predictive vectors over each factor separately, in orthogonal subspaces (Figure 1a, bottom right). This requires only $\sum_n (d_n - 1)$ dimensions—exponentially fewer (Figure 1c), but sacrifices fidelity when factors aren't conditionally independent (Figure 1b).

For the task of predicting future tokens, the factored representation is sufficient when a key property holds: when the generator is conditionally independent, latent states that factorize across parts remain factored through context. More precisely:

**Proposition 2.2.** *When the transition operators are conditionally independent, the predictive vector remains a* product state*:*

$$\boldsymbol{\eta}^{(x_{1:\ell})} = \bigotimes_{n=1}^{N} \boldsymbol{\eta}_n^{(x_{1:\ell})} \implies \boldsymbol{\eta}^{(x_{1:\ell+1})} = \bigotimes_{n=1}^{N} \boldsymbol{\eta}_n^{(x_{1:\ell+1})}. \quad (5)$$

Here, *product state* refers to a special class of predictive vectors in $\mathbb{R}^{\prod_n d_n}$ that can be expressed as a tensor product over the state of each factor. These product states form a low-dimensional sub-manifold within the full joint space (bottom left of Figure 1a), and conditionally independent dynamics keep beliefs on this sub-manifold.[3] Importantly, this implies that, when the token-induced transition dynamic is conditionally independent, the relevant predictive vectors can be losslessly represented in factored form (bottom left of Figure 1a, App. A.4 for details):

**Theorem 2.3.** *There exists a lossless linear map from this product-state sub-manifold to a factored tensor product representation where the local predictive vectors for each subspace live in orthogonal subspaces:*

$$\boldsymbol{\eta}_{FWH}^{(x_{1:\ell})} \equiv \bigoplus_{n=1}^{N} \boldsymbol{\eta}_n^{(x_{1:\ell})} . \quad (6)$$

The map highlighted in the theorem is invertible on product states: no predictive information is lost by representing factors in orthogonal subspaces rather than the exponentially larger joint space. However, when factors are *not* conditionally independent (i.e., when observations induce correlations that cannot be factored) the predictive vector moves off the product-state manifold (Figure 1b). The factored representation then becomes lossy: projecting onto it

---

[3]More precisely, the product-state sub-manifold maps to itself under conditionally independent generators, and the projective normalization in belief updating is contractive, so even beliefs that start slightly correlated will converge toward factored states.

discards information about cross-factor correlations, degrading predictions. This creates a tradeoff, as the joint representation is always faithful but expensive, and the factored representation is cheap but only lossless when factorization is valid.

The experimental results presented in Section 3 show that transformers learn factored representations when this is consistent with the training data, and learn them early in training *even when the data is not entirely consistent with this factored representation.*

### 2.5. The Factored World Hypothesis

The analysis above establishes that factored representations are *possible* when the data-generating process admits conditional independence, but possibility does not imply preference. A transformer with sufficient capacity could represent the full joint space regardless. The question is whether transformers *choose* to factor.

> **Factored World Hypothesis (FWH)**
>
> Transformers have an inductive bias toward factored representations, organizing their activations into a direct sum of low-dimensional orthogonal subspaces even when higher-dimensional representations would be more faithful.

This hypothesis makes strong empirical commitments. It is not enough for the factored representation to be *recoverable* from activations via some nonlinear transformation; we claim it is the *native* representation, accessible via linear projection. And we claim this holds even when model capacity would easily accommodate the joint alternative.

### 2.6. Testable predictions

The factored world hypothesis yields the following testable predictions:

1. **Dimension.** If the data-generating process has $N$ conditionally independent factors with latent dimensions $d_1, \ldots, d_N$, activations should concentrate in $\sum_n (d_n - 1)$ dimensions, not $(\prod_n d_n) - 1$.

2. **Orthogonality.** These dimensions should organize into $N$ orthogonal subspaces, one per factor.

3. **Preference for factoring.** These properties should hold even when model capacity could accommodate the full joint representation, and in cases where the data generator does not perfectly factor. The factored solution is *preferred*, not merely sufficient.

# 3. Experiments

To test whether transformers learn representations that are factored, we train networks on sequence data emitted by generators with different types of factorizable structure. Our experiments directly test the predictions from Section 2.6. We construct data-generating processes where the ground-truth joint and factored representations are analytically computable, then probe whether trained models learn one, the other, or something in between.

**Factors and token construction.** In all experiments, we compose five elementary generators as factors. In each experiment, we use three **Mess3** factors (3-state HMMs, each emitting subtokens from a ternary alphabet $\mathcal{Z}_n = \{0, 1, 2\}$ for $n \in \{1, 2, 3\}$) and two **Bloch Walk** factors (3-dimensional GHMMs, each emitting subtokens from a quaternary alphabet $\mathcal{Z}_n = \{0, 1, 2, 3\}$ for $n \in \{4, 5\}$). We choose these elementary factors because their individual predictive-state geometries are already well understood from prior **activation geometry** work (Shai et al., 2024; Riechers et al., 2025). When trained on each process independently, transformers linearly recover the corresponding Mess3 and Bloch Walk geometries. Here, however, the model is trained only on a single observed integer token formed by mapping the Cartesian product of the subtokens $(z_\ell^{(1)}, \ldots, z_\ell^{(5)})$ to integers: $x_\ell \in \mathcal{X} = \{\text{BOS}, 0, 1, \ldots, 431\}$, giving a vocabulary of size $|\mathcal{X}| = 3^3 \times 4^2 + 1 = 433$ (including BOS). Importantly, the model sees only these integer tokens and must discover any factored structure from the standard pretraining on next-token cross-entropy loss. This experiment is designed to test whether the network represents the resulting predictive state as one high-dimensional joint geometry, or recovers the constituent factor geometries in separate subspaces. Both are valid solutions consistent with the ground truth data structure.

**Scope.** In our synthetic generators, the predictive vector for each factor induced by a context is in one-to-one correspondence with the factor's next-subtoken distribution. Our experiments test how transformers represent this predictive information—specifically, whether it is encoded in a joint (tensor product) geometry or in a factorized (direct sum) geometry. Our focus is the geometric signature of this representational choice: factored or joint.

**Experimental regimes.** We consider three regimes that progressively stress-test the degree to which transformer representations have an inductive bias towards factorization:

1. **Independent factors** (Section 4.1): Five processes running in parallel with transition operators $T^{(x)} = \bigotimes_{n=1}^{N} T_n^{(z^{(n)})}$. The joint representation varies in 242 dimensions, while the factored varies in only 10.

2. **Conditionally independent factors** (Section 4.2): A dependency chain where each factor's dynamics depend on preceding factors' outputs, but tensor product structure is preserved: $T^{(x)} = \bigotimes_{n=1}^{N} T_n^{(z^{(n)}|z^{(1:n-1)})}$. In contrast to the previous case, here there is covariance structure amongst the states of the factors.

3. **Indecomposable generators** (Section 4.3): In this case we create a situation where the transformer must choose between a dimensionally efficient but lossy factored representation and high-dimensional but optimal joint representation. To do so we use transitions $\varepsilon$-close to factored form, $T^{(x)} = \varepsilon T_{\text{int}} + (1 - \varepsilon) \bigotimes_{n=1}^{N} T_n^{(x)}$, where small $\varepsilon$ keeps predictive vectors near the product-state manifold while large $\varepsilon$ pushes them off (Figure 1b). This creates tension between dimensional efficiency and predictive fidelity.

**Architecture and training.** All experiments use a GPT-2 style decoder-only transformer trained with standard next-token cross-entropy loss. We train models with capacity both above and below what the full joint geometry requires; in all cases the model retains enough capacity for the factored geometry. The figures in the main text use models that have 4 layers, model dimension $d_{\text{model}} = 120$, and MLP dimension $d_{\text{MLP}} = 480$. We also trained on larger models, with $d_{\text{model}} = 480$, and $d_{\text{MLP}} = 1920$, with qualitatively similar results shown in Appendix D.3. We use the Adam optimizer (Kingma & Ba, 2014) with no weight decay. Training sequences have length $L = 8$. We find similar results for $L = 33$ and $L = 101$, which we show in Appendix D.3. For further training details see Appendix E. Full hyperparameters and code are available here.

# 4. Experimental results

Our theoretical framework makes precise predictions about the geometry of transformer activations, including the number of dimensions used, their organization into orthogonal subspaces, and the arrangement of context embeddings within each subspace. We now test these predictions systematically, beginning with the simplest case and progressively introducing complications that stress-test the degree to which transformers have an inductive bias towards factored representation.

## 4.1. Transformers learn independent factors

We first consider the case of five factors running in complete independence, as described above. Each factor emits subtokens that combine into a single observed integer token. The transformer sees only these integer tokens, and nothing in the experimental setup signals to the network the existence of five underlying factors.

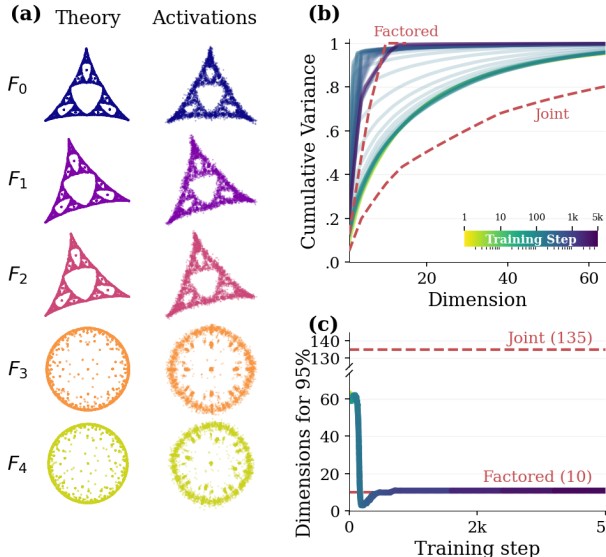

*Figure 2.* **Transformers discover latent factored structure.** We train a transformer on sequences generated by five independent factors (F0–F4); the model sees only tokens corresponding to the Cartesian product of factor outputs. (a) Left: ground-truth belief geometry for each factor. Right: linear regression from activations recovers each geometry with high fidelity. (b) Cumulative explained variance (CEV) over training. Dashed lines show predictions for factored versus joint representations; the model converges to the factored prediction. (c) Effective dimensionality (dimensions for 95% variance) collapses to about 10 during training, far below the joint requirement. Dashed lines show dimensions required to explain 95% of the variance for the factored (10 at 95% CEV) and joint (135 at 95% CEV) predictions.

Each factor has a 3-dimensional latent space. The joint latent space is the tensor product of these spaces, giving dimension $3^5 = 243$. Since predictive vectors are normalized over this joint space, they vary in $243 - 1 = 242$ dimensions. In contrast, the factored representation tracks each factor's predictive vector separately. Each of these is also normalized, so each factor varies in $3 - 1 = 2$ dimensions, giving $5 \times 2 = 10$ total dimensions.

**Recovering factor geometry.** We first verify that the transformer has a representation of the predictive vectors at all. For each factor $n$, we fit a linear regression from residual stream activations to the ground-truth predictive vectors $\boldsymbol{\eta}_n^{(x_{1:\ell})}$. Figure 2(a) shows that all five factor geometries are recovered with high fidelity, with RMSE decreasing monotonically over training alongside cross-entropy loss, shown in Appendix D.1. This shows activations contain information corresponding to the latent state of each factor, but leaves open whether the model represents these factors jointly or separately.

**Dimensionality of Representations** The two representations make sharply different predictions about how activa-

tions are spread across the dimensions in the residual stream of the transformer, shown as dashed lines in Figure 2b. Colored lines show the cumulative explained variance (CEV) of residual stream activations over training. At initialization, the activations are spread across the residual stream dimensions, with ∼100 dimensions required to explain 95% of the variance (Figure 2c). Early in training, activations compress to below the number of dimensions required for the factored representations. Shortly after, the activations change their arrangement to more closely align with the prediction of the factor representation. In addition, the number of dimensions required to explain 95% of the variance plateaus at 11 dimensions, closely matching the factored prediction and ruling out the joint alternative. We verify these results hold on Transformers with $d_{\text{model}}$ up to 480, showing a strong preference for factoring even when the residual stream has abundant capacity (Appendix D.3).

**Orthogonality of factor subspaces.** The factored world hypothesis predicts that each factor's information lives in its own linear subspace, and that these subspaces are mutually orthogonal. To test this, we use a "vary-one" analysis (outlined in Appendix H.1.1) to first identify candidate subspaces. We compare these subspaces using two complementary orthogonality analyses to explore their relative organization in activation space (outlined in Appendix H.2).

The vary-one procedure is as follows: for each factor, we construct datasets where only that factor's subtoken sequences vary, keeping the subtoken sequences of the other four fixed. For each fixed configuration of the non-varied factors, we collect and mean-center the activations from an ensemble of realizations of the varied factor. We then aggregate across many configurations and perform PCA. The top $k$ principal components define the $k$-dimensional candidate subspace for that factor.

In Figure 3(a) we explore the geometric arrangement of these subspaces over training steps by tracking the effective dimensionality (at 95% variance explained) for each factor's vary-one dataset, as well as the effective dimensionality of all vary-one datasets aggregated (labeled "Union"). At initialization, the sum of individual factor dimensionalities far exceeds the union, so factors must share directions (see Appendix H.2.1). Over training, the sum falls to 14—close to the union's dimension of 12, indicating approximate but not exact orthogonality.

Figure 3(b) offers a complementary perspective by directly quantifying overlap between factor subspaces. We use a normalized subspace overlap metric that ranges from 0 (perfectly orthogonal) to 1 (fully overlapping), detailed in Appendix H.2.2. For each factor, we take the $k$-dimensional candidate subspace resulting from the vary-one analysis and compute pairwise overlaps across all other factor subspaces.

The figure shows these pairwise overlaps at various training steps (colored lines), and the average at each step.

At initialization, factor subspaces overlap substantially regardless of $k$. Over training, overlaps drop markedly across all ten factor pairs, but only in the first two principal components. This is consistent with the transformer representing each factor in an approximately two-dimensional orthogonal subspace. Projecting activations onto these subspaces yields the anticipated predictive vector geometry while higher-order principal components capture relatively little variance, these results are shown in Appendix H.1.1.

**Factorization begins at the embedding.** Recall that the model sees only integer tokens, with nothing in the training signal indicating that each token implicitly encodes subtokens from five independent alphabets. Remarkably, after training, factored structure emerges at the token embedding, before any transformer blocks. The learned embedding matrix has only $\sim$12 non-negligible singular values, corresponding to the direct sum of subtoken simplices (three 2-simplices for the Mess3 factors plus two 3-simplices for the Bloch Walk factors, Figure 14). Successive principal components partition tokens by subtoken identity for each factor. This structure is absent at initialization—the model discovers the latent subtoken decomposition and assigns orthogonal subspaces to each factor at the earliest possible point (Appendix D.6).

## 4.2. Transformers learn conditionally independent factors

The condition for exact factoring permits dependencies among the factors, provided their joint evolution can be described by Eq. (4). These processes include all directed acyclic graphs (DAG) of dependencies between factors (Boyd et al., 2025), with hierarchical structure among latent factors as a prominent example. To investigate whether factoring happens empirically in this broader setting, we train a transformer on data generated from a chain of three Mess3 processes and two Bloch Walk processes, similar to our earlier experiments—but now each of the subprocesses depends on all of the factors above it in the chain.

We find that neural networks trained via next-token prediction on the resulting sequences automatically learn to factor their representation into parts, where uncertainty over the latent states of each part is encoded geometrically. See Appendix D.2 for additional details.

## 4.3. Factored representations as an inductive bias

The previous experiments demonstrate that transformers learn factored representations when factoring is valid. But this leaves open a crucial question: is factoring merely a sufficient solution that the model stumbles upon, or is it a

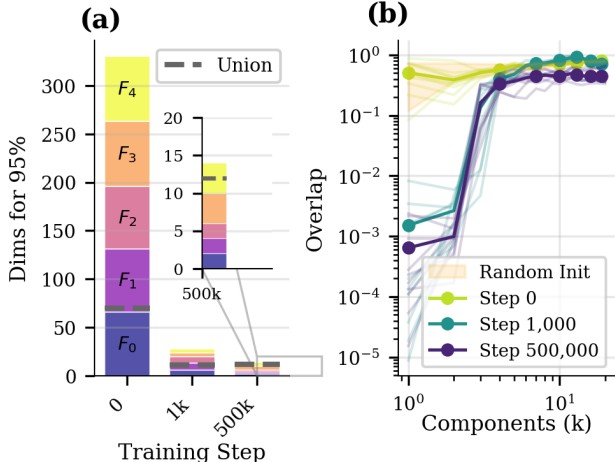

*Figure 3.* **Transformers represent factors in orthogonal multi-dimensional subspaces.** (a) The number of dimensions needed to reach a CEV of 95% in the residual stream at the output layer of the last transformer block when driven by data from the vary-one dataset. At initialization, the activations associated with each factor use approximately the same number of dimensions as the union suggesting the subspaces are effectively fully overlapping. Over training the sum of the factor's dimensions approaches the dimension of the union, which suggests orthogonality. (b) Indeed, the first two principal components (PCs) associated with each factor evolve over training to become orthogonal to those associated with other factors. Thin lines show the normalized subspace overlap between the PCs belonging to different factors, while the heavy lines are the average over all pairs at each training step. The highlight shows a 90% confidence interval for initial factor overlap, generated from an ensemble of randomly initialized models.

preferred solution that the architecture actively seeks?

To distinguish these possibilities, we need to examine cases where factoring is *not* valid—where the factored representation *is* lossy, and the model must choose between representing the world as made of parts and predictive fidelity.

**Controlled violations of conditional independence.** We take our five independent factors and pass the token through a noisy channel. With probability $1 - \varepsilon$, the token is transmitted faithfully; with probability $\varepsilon$, it is replaced by a uniformly random token. For example, at $\varepsilon = 0.5$, there is a 50% chance that the observed token reflects the outputs of the five factors, and a 50% chance it is pure noise. This simple corruption breaks conditional independence. When the channel transmits faithfully, observing the token updates information about each factor independently. But when the channel corrupts, the observation provides no information, and critically, the observer doesn't know which case occurred. The result is that beliefs about different factors become correlated in ways that cannot be factored. Predictive vectors drift off the product-state manifold, and projecting onto the factored representation discards relevant information about these cross-factor correlations.

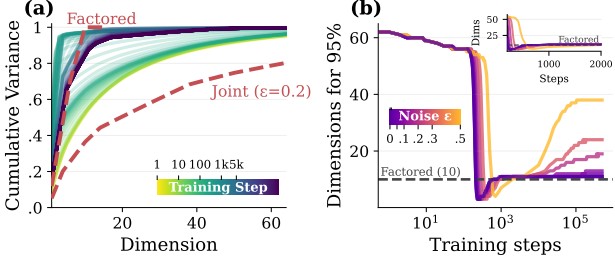

Figure 4. **Transformers have an inductive bias towards factored representations.** Even when the generative process doesn't admit a lossless factorization (as is the case for nonzero values of $\varepsilon$), the transformer quickly learns one anyways. For small $\varepsilon$, this factored representation persists throughout training. As $\varepsilon$ grows, transformers find the factored form early before expanding their dimensionality for greater predictive fidelity.

This creates a measurable tradeoff via a loss gap between the factored and the full representations that we quantify in Appendix C.4. In short: the factored representation offers dimensional efficiency at the cost of predictive fidelity, while the joint representation offers fidelity at the cost of dimensionality. A model that treats factoring as merely sufficient would use whatever representation minimizes loss. A model with an inductive bias toward factoring would prefer the factored solution even when it hurts.

**Explained Variance over Training.** Figure 4(a) shows the CEV curves over training for $\varepsilon = 0.2$, when the noisy channel has a 20% chance of randomizing the hidden tokens (See Figure 13 for other $\varepsilon$ analysis). Over training, activations compress and converge close to the dimensionality of the factored representations we find in Figures 2 and 3. Notably, this happens despite the fact that the factored representation is lossy.

**Factored reps are preferred even when non-optimal.** Figure 4(b) shows the number of principal components required to capture 95% of activation variance over training, for training runs at several values of $\varepsilon$. Remarkably, regardless of $\varepsilon$, transformers first collapse to the 10-dimensional factored representation within hundreds of gradient steps. Only later, under sustained pressure from the loss gradient, do they gradually expand into higher dimensions.

This is not what we would expect if the model were simply finding the lowest-loss solution. At high $\varepsilon$, the factored representation leaves substantial predictive information on the table, information the model demonstrably learns to recover later in training. Yet the model dwells in the factored regime first, departing only reluctantly.

The factored solution acts as a representational attractor. Models find it quickly, dwell there, and depart only under sustained gradient pressure. This suggests that factorization

is not merely a capacity-saving trick for tractable problems, but a genuine inductive bias of the architecture.

We note that the strength of this inductive bias is not, as of yet, well quantified. For example, whether these models still try to factor when they lack even the capacity to hold the full factored representation is an intriguing open question for future development. This question can be directly addressed with an architecture-specific mechanistic picture of factoring that identifies where the architectural bottlenecks lie.

### 4.4. RNNs also learn to factor

We ran the same experiments and analysis on vanilla RNNs and LSTMs to check whether our results are particular to transformers. We find that they partially generalize: in all cases, we find CEV curves that are more consistent with a factored representation than a joint one, though with the precise shape varying based on $d_{\text{model}}$. All tested models contain linear maps from their activations to all 5 factors. For RNNs, at lower $d_{\text{model}}$, we find that the CEV curves closely match our Transformer results, with 10 components explaining 95% of the cumulative variance. As we scale up $d_{\text{model}}$, the CEV curves smooth out even when the model does not have capacity for the joint representation. For LSTMs, we observe this phenomenon at smaller $d_{\text{model}}$ sizes. See Appendix. D.4 for more details.

### 5. Related work

That neural networks learn structured, factored, or distributed representations is a longstanding observation, dating at least to Hinton's family-trees model (Hinton, 1986), in which an explicit bottleneck pressured the network to discover factored codes. Closest to our setting, Trager et al. (2023) establish, in their Corollary 8, that embeddings of a multimodal model factor linearly across latent attributes if and only if those attributes are conditionally independent given the observation. We extend this structural result into the autoregressive next-token setting, derive a competing geometric prediction, and stress-test in the regime where conditional independence is violated. Concurrent work by Noroozizadeh et al. (2026) likewise reports that geometric representational structure arises without any capacity bottleneck, though their geometry tracks graph topology of entity embeddings rather than the tensor product factorization of predictive states we identify here.

Against this backdrop, our contribution is not solely the observation *that* pretrained networks acquire structured representations, but a theoretical framework that predicts *which* geometry should be acquired. This lets us distinguish factoring-as-optimality (when factors are conditionally independent) from factoring-as-inductive-bias (when the factored solution is provably lossy). Our experiments

confirm the latter: a phenomenon the prior literature on structured-representation emergence does not anticipate.

There have been many efforts to *engineer* factored representations (Degris & Sigaud, 2013; Sallans, 1999). For example, RL has been used to learn factored Markov decision processes (Guestrin et al., 2003; Strehl et al., 2007), and much work has gone into learning the structure of dynamic Bayesian networks (Kaddour et al., 2022). In contrast, our results investigate how factorization *naturally* arises through pretraining. Specifically, we observe that neural networks discover relevant factored belief representations without any explicit network, graph, or reinforcement learning.

New architectures have been proposed to force factored representations (Dinh et al., 2014; Nanbo et al., 2025). In our experiments, we find that neural networks do this during next-token pretraining, with no new architecture needed.

In the field of interpretability, our work is consistent with recent work that also finds multi-dimensional geometric relationships among activations in LLMs (Engels et al., 2025; Gurnee et al., 2025; Kantamneni & Tegmark, 2025). A novel contribution of our work is its constructive nature: we predict what these higher dimensional features *should* be. These multi-dimensional belief geometries are dense in their active subspaces, which may help explain why some sparse-autoencoder (SAE) latents are dense (Sun et al., 2025). This invalidates the explicit assumption motivating SAEs (Elhage et al., 2022), that all concepts would be orthogonal if given sufficiently many dimensions. We find that nonorthogonality of activations within the factor's subspace purposefully encodes similar predictions. This may enable some ability to anticipate and interpret potentially universal context embeddings (Huh et al., 2024; Sucholutsky et al., 2023).

When representations factor into orthogonal subspaces, it allows an understanding of which behaviors can be tuned independently, enabling rather surgical interventions (Feucht et al., 2025). It also hints at how fine tuning on one aspect of behavior in one context can enhance or suppress that behavior across others (Betley et al., 2025).

The emergence of belief geometry from pretraining was first found by Shai et al. (2024), and its mechanisms are just beginning to be understood (Piotrowski et al., 2025; Riechers et al., 2025). The present contribution shows that factoring beliefs into conditionally independent parts allows further dimensionality reduction in representations of Bayesian beliefs over generative world models.

Boyd et al. (2025) describe the information theory of factorizing processes into modular subcomponents, showing how the composition of stochastic automata (Hopcroft & Ullman, 1979; Mohri, 1997; Mohri et al., 2002) can be inverted. Our work suggests neural networks automatically learn a representational version of this causal graph discovery (Nogueira

et al., 2022), similar to transformers' ability to learn Krohn–Rhodes and other decompositions to efficiently simulate deterministic automata (Liu et al., 2023).

Our findings offer some reason for representational optimism, in tension with the fractured entangled representation hypothesis of Kumar et al. (2025); we identify a setting in which task structure itself supplies the inductive bias that Locatello et al. (2019) proved necessary for disentanglement.

## 6. Conclusion

In this work, we discovered that, as a natural consequence of next-token pretraining, neural networks automatically learn to factor the world into parts. To make rigorous falsifiable claims, we trained relatively small transformers on various sources of correlated sequences designed to test when and how these models factor their activations.

We found that we could precisely anticipate the learned representations—how many dimensions of the residual stream would be used, how many multi-dimensional subspaces would be updated in parallel, and even the precise geometric pattern of non-orthogonal activations induced by different contexts within each subspace. We found an inductive bias for transformers to factor their internal activations early in training, even when this sacrifices predictive fidelity. We ran analogous experiments on RNN architectures, and found that they too learn these factored representations, hinting at the universality of these results.

Given the theoretical underpinnings of our empirical results, it is plausible that the factored world hypothesis extrapolates to larger models as well. [4] This sets the stage for many new opportunities in interpretability: rather than needing to understand the entire neural network all at once, this gives hope that we can discover some low-dimensional interpretable subspaces in the residual stream. New interpretability tools will be needed to discover these factored-representation subspaces in an unsupervised fashion.

We could then trace information flow within and amongst these subspaces, each a separable part of the cognitive whole. With this computational universe unentangled into its basic constituents, we may then hope to build a fundamental psychology, respecting the structure of concepts that models natively represent.

---

[4]Indeed, for frontier models trained on such massive datasets, there will be extreme pressure to factor the learned world model into parts since this yields the exponential dimensional advantage that the model would need to capture much of anything in the latent space of its finite model dimension.

## Acknowledgments

We thank Eric Michaud, Javan Tahir, Xavier Poncini, Matt Levinson, and Jake Ward for helpful comments and discussion.

## Impact Statement

This paper presents work whose goal is to advance the field of interpretability. There are many potential societal consequences of our work, none which we feel must be specifically highlighted here.

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

# A. Mathematical details

## A.1. GHMMs

We consider non-Markovian sequence data generated by finite-dimensional generalized hidden Markov models (GHMMs)—a general class of models with latent transition operators[5] that includes as special cases (i) hidden Markov models (HMMs), and (ii) quantum hidden Markov models (QHMMs) (Riechers & Elliott, 2025). The latter describes repeated evolution and interaction with quantum density matrices, which is arguably the most general type of data-generating mechanism possible.[6]

A GHMM is defined by the four-tuple

$$\mathcal{M} = \left( \mathcal{X}, \mathcal{S}, \boldsymbol{\eta}^{(\varnothing)}, (T^{(x)})_{x \in \mathcal{X}} \right), \tag{7}$$

where $\mathcal{X}$ is the token alphabet, $\mathcal{S} = \mathbb{R}^d$ is a real vector space that serves as the latent space of the GHMM, $\boldsymbol{\eta}^{(\varnothing)}$ is the initial row vector in $\mathcal{S}$, and the net transition operator $T = \sum_{x \in \mathcal{X}} T^{(x)}$ has an eigenvalue of unity with associated right eigenvector $\mathbf{1} = T\mathbf{1}$, normalized such that $\boldsymbol{\eta}^{(\varnothing)}\mathbf{1} = 1$. The associated left eigenvector $\boldsymbol{\pi} = \boldsymbol{\pi}T$, also normalized such that $\boldsymbol{\pi}\mathbf{1} = 1$, is the *steady-state* of the GHMM. The ground-truth probability of any sequence $x_{1:\ell} \in \mathcal{X}^\ell$ of arbitrary length $\ell$ can be calculated directly via linear algebra:

$$Q_{\mathcal{M}}(X_{1:\ell} = x_{1:\ell}) = \boldsymbol{\eta}^{(\varnothing)}T^{(x_{1:\ell})}\mathbf{1} , \tag{8}$$

where $T^{(x_{1:\ell})}$ is simply obtained by matrix multiplication $T^{(x_{1:\ell})} = T^{(x_1)} \cdots T^{(x_\ell)}$.

HMMs are a special case of GHMMs. The matrix elements of an HMM correspond to joint conditional transition probabilities $T_{s,s'}^{(x)} = Q(s', x | s)$ of observing token $x$ and moving to state $s'$ when starting in the hidden state $s$. In this special case, $\boldsymbol{\eta}^{(\varnothing)}$ is a probability distribution over the hidden states, and $\mathbf{1}$ is the column vector of all ones.

## A.2. Belief updating implies geometry

The conditional probability density over the future $\overrightarrow{X}$ of any length, induced by some context $\overleftarrow{x}$, can be written in terms of the linear algebra of the GHMM as

$$Q(\overrightarrow{X} | \overleftarrow{x}) = \frac{\boldsymbol{\eta}^{(\varnothing)}T^{(\overleftarrow{x})}}{\boldsymbol{\eta}^{(\varnothing)}T^{(\overleftarrow{x})}\mathbf{1}}T^{(\overrightarrow{X})}\mathbf{1} = \boldsymbol{\eta}^{(\overleftarrow{x})}T^{(\overrightarrow{X})}\mathbf{1} . \tag{9}$$

This immediately highlights the important role of the *predictive vector*

$$\boldsymbol{\eta}^{(\overleftarrow{x})} := \frac{\boldsymbol{\eta}^{(\varnothing)}T^{(\overleftarrow{x})}}{\boldsymbol{\eta}^{(\varnothing)}T^{(\overleftarrow{x})}\mathbf{1}} , \tag{10}$$

and implies that, whenever two contexts $\overleftarrow{x}$ and $\overleftarrow{x}'$ induce the same predictive vector $\boldsymbol{\eta}^{(\overleftarrow{x})} = \boldsymbol{\eta}^{(\overleftarrow{x}')}$, these two contexts cannot affect the future differently. More generally, when two contexts $\overleftarrow{x}$ and $\overleftarrow{x}'$ induce nearby predictive vectors, these two contexts must make similar predictions about the future.

Altogether, the collection of these predictive vectors creates the *belief geometry* associated with the correlational structure of the sequence data. This belief geometry emerges in the activations of neural networks as they are pretrained on next-token prediction (Shai et al., 2024; Piotrowski et al., 2025; Riechers et al., 2025); during inference, the expected predictive vectors are induced as the model moves through context. Nonorthogonality of these vectors implies similar predictions induced by these contexts in the overlapping directions they share in activation space.

---

[5]The linearity of latent transition operators may seem limiting, yet these GHMMs can describe observations of paradigmatically nonlinear complex systems. There are two ways of seeing that the linearity in the latent space is sufficient to describe all systems: 1) Probability densities always map linearly according to the Ruelle–Perron–Frobenius operator, even for chaotic systems like the beloved Lorenz system. 2) Observations always map linearly according to the Koopman operator. These views are dual, in the linear algebraic sense.

[6]Although see Fanizza et al. (2024) and Riechers et al. (2025), which point out that generalized probabilistic theories (Plávala, 2023) allow for more compact representations, which are in fact still GHMMs.

### A.3. Joint representation and the product-state submanifold

For a system composed of $N$ parts—with the $n^{\text{th}}$ part having a local $d_n$-dimensional latent space $\mathbb{R}^{d_n}$—the most general predictive vector lives in a $((\prod_{n=1}^N d_n) - 1)$-dimensional affine subspace of $\mathbb{R}^{\prod_{n=1}^N d_n}$. We can also refer to such a general predictive vector as a *joint state*. For a classical system with a fixed interpretable orthonormal basis for the latent space, each predictive vector corresponds to a joint probability distribution over the composite latent states of all parts, and lives in the $((\prod_{n=1}^N d_n) - 1)$-dimensional probability simplex over joint latent states.

In generalized probabilistic theories like quantum mechanics (Plávala, 2023), it is always possible to choose a real vector space to represent the predictive states (e.g., generalized Bloch coordinates to represent the density matrix). In this case, there may not be a distinguished latent basis, but the predictive vectors will nevertheless all live in a convex space. For a single quantum system, this could be the Bloch ball for a qubit, or the generalized Bloch ball for larger quantum systems. These nominally post-classical systems allow for rather low-dimensional representations of the correlational structure of token sequences (Fanizza et al., 2024), and it has been shown that transformers use the real vector space of their residual stream to take advantage of these compressed representations (Riechers et al., 2025).

Following the terminology of multipartite quantum mechanics, a **product state** is a joint state that is equal to the tensor product of its reduced marginals. I.e., product states are of the form $\boldsymbol{\eta} = \bigotimes_{n=1}^N \boldsymbol{\eta}_n$. The set of all product states forms the **product-state manifold**—a curved low-dimensional manifold in the joint space. Since each reduced state $\boldsymbol{\eta}_n$ has only $d_n - 1$ degrees of freedom, the product-state manifold has an intrinsic dimension of only $\sum_{n=1}^N (d_n - 1)$ dimensions, although its linear span is the full $(\prod_{n=1}^N d_n)$-dimensional joint space.

Joint states that are off the product-state manifold encode correlations among parts. Sec. A.5 shows both (i) how to marginalize over other parts of the joint to obtain the *reduced state* of a subsystem, and (ii) how to quantify the *total correlation* among parts. In general, projecting a state onto the product-state manifold—by taking the tensor product of its reduced states—throws away information about how the parts are correlated, and thus compromises forecasts about the future. However, sometimes parts really are independent, or observations render the parts conditionally independent.

Recall that a GHMM generator is *conditionally independent* if each token-labeled linear operator can be expressed as the tensor product of multiple linear operator factors: $T^{(x)} = \bigotimes_{n=1}^N T_n^{(x)}$. These conditionally independent operators map the product-state manifold to itself since $\left(\bigotimes_{n=1}^N \boldsymbol{\eta}_n\right)\left(\bigotimes_{n=1}^N T_n^{(x)}\right) = \bigotimes_{n=1}^N \boldsymbol{\eta}_n T_n^{(x)}$.

Accordingly, when (i) the initial predictive vector is a product state $\boldsymbol{\eta}^{(\varnothing)} = \bigotimes_{n=1}^N \boldsymbol{\eta}_n^{(\varnothing)}$ and (ii) the token-labeled latent transition operators are all of the same product form $T^{(x)} = \bigotimes_{n=1}^N T_n^{(x)}$, the belief updates maintain a tensor product structure of the predictive vector[7] such that

$$\boldsymbol{\eta}^{(x_{1:\ell})} = \bigotimes_{n=1}^N \boldsymbol{\eta}_n^{(x_{1:\ell})} \overset{x_{\ell+1}}{\mapsto} \boldsymbol{\eta}^{(x_{1:\ell+1})} = \bigotimes_{n=1}^N \boldsymbol{\eta}_n^{(x_{1:\ell+1})} \tag{11}$$

for all $\ell$ and all $x_{1:\ell+1} \in \mathcal{X}^{\ell+1}$, then optimal forecasts about the future can be fully described via the collection of beliefs over each factor locally.[8] In this case, we say that the factors are *conditionally independent* given the context—although they are notably *not* necessarily independent if we marginalized over historical context.

These predictive vectors in Eq. (11) lie on the curved product-state manifold of $\sum_{n=1}^N \left(\dim(\boldsymbol{\mathcal{S}}_n) - 1\right)$ dimensions[9] within the larger tensor product space.[10] Crucially, the dimension of this submanifold grows only linearly in the number of parts, although the dimension of the tensor product space grows exponentially in the number of parts.

---

[7]in a fixed tensor product decomposition of the vector space

[8]Even if the initial predictive vector is not a product state, the predictive vector will generically approach the product-state submanifold upon repeated application of product-preserving maps.

[9]The "$-1$" here comes from the fact that each constituent predictive vector $\boldsymbol{\eta}_n$ lives in an affine hyperplane of dimension $\dim(\boldsymbol{\mathcal{S}}_n) - 1$, since differences in these local predictive vectors are orthogonal to $\mathbf{1}_n = \left(\sum_{z^{(n)}} T_n^{(z^{(n)}|z^{(1:n-1)})}\right)\mathbf{1}_n$. If the sub-process is a (possibly input-dependent) HMM, then these local predictive vectors are belief states that live in the $\left(\dim(\boldsymbol{\mathcal{S}}_n) - 1\right)$-simplex over the HMM's hidden states.

[10]This is essentially the Segre variety from projective geometry.

## A.4. Factored representations

> **Key insight: Product states can be represented as a collection of predictions in orthogonal subspaces**
>
> There is a lossless linear projection from the submanifold of tensor product states—represented in the joint predictive state space—to an exponentially lower number of dimensions where the beliefs over the parts are represented in orthogonal subspaces.

Although the linear span of all product states has dimension equal to the full joint space $\prod_{n=1}^{N} \dim(\boldsymbol{S}_n)$, the submanifold containing all product states is of much lower dimension $\sum_{n=1}^{N}\big(\dim(\boldsymbol{S}_n) - 1\big)$. There is a linear map from the joint space to a much lower-dimensional factored space. This mapping is invertible when the domain of the map is limited to the product states. See Fig. 1.

In the case of conditionally independent factors, each *local* predictive vector $\boldsymbol{\eta}_n^{(w)}$ can be updated locally in parallel via

$$\boldsymbol{\eta}_n^{(w)} \overset{x}{\mapsto} \boldsymbol{\eta}_n^{(wx)} = \frac{\boldsymbol{\eta}_n^{(w)} T_n^{(x)}}{\boldsymbol{\eta}_n^{(w)} T_n^{(x)} \mathbf{1}_n}. \tag{12}$$

Rather than directly representing $\boldsymbol{\eta}^{(w)}$ in $\prod_{n=1}^{N} \dim(\boldsymbol{S}_n)$ dimensions, we expect the model to directly represent

$$\boldsymbol{\eta}_{\mathrm{FWH}}^{(w)} = (\tilde{\boldsymbol{\eta}}_1^{(w)}, \tilde{\boldsymbol{\eta}}_2^{(w)}, \ldots, \tilde{\boldsymbol{\eta}}_N^{(w)}) = \sum_{n=1}^{N} \iota_n(\boldsymbol{\eta}_n^{(w)}) \tag{13}$$

using only $d_{\mathrm{FWH}} = \sum_{n=1}^{N}\big(\dim(\boldsymbol{S}_n) - 1\big)$ dimensions, where $\iota_n : \mathbb{R}^{d_n} \to \mathbb{R}^{\sum_{n'=1}^{N}(d_{n'} - 1)}$ linearly embeds each local predictive vector into the same $d_{\mathrm{FWH}}$-dimensional embedding space. This is effectively just concatenating the predictive degrees of freedom $\tilde{\boldsymbol{\eta}}_n^{(w)} \in \mathbb{R}^{d_n - 1}$—which is $\boldsymbol{\eta}_n^{(w)}$ projected onto the subspace orthogonal to $\mathbf{1}_n$—for each subsystem.

In the main text, we write this less formally (but hopefully more intuitively) via the direct sum $\bigoplus$ (a notation usually reserved for vector spaces rather than vectors themselves). Context embeddings in the factored representation live in the vector space obtained from the direct sum of constituent vector spaces for each part. This corresponds to adding the vectors for each component only after they've been embedded orthogonally into this shared space.

In particular,[11] the linear embedding map takes the form $\iota_n = \sum_{m=2}^{|\mathcal{B}_n|} |b_{m,n}\rangle\langle\delta_{m-1+\sum_{n'=1}^{n-1}(d_{n'}-1)}|$, where $\langle\delta_k|$ is a row vector which is one-hot at its $k^{\mathrm{th}}$ element. $\mathcal{B}_n = \{|b_{m,n}\rangle\}_{m=1}^{|\mathcal{B}_n|}$ is a set of $|\mathcal{B}_n|$ orthonormal vectors with $|b_{1,n}\rangle = |\mathbf{1}_n\rangle$. The other basis elements $\{|b_{m,n}\rangle\}_{m=2}^{|\mathcal{B}_n|}$ are rather arbitrary (besides orthogonality with $\mathbf{1}_n$) but can be obtained by Gram–Schmidt orthogonalization (or PCA on the mean-centered local predictive vectors).

In effect, the factored representation linearly embeds each local predictive vector, allotting one dimension to each degree of freedom for each predictive vector. I.e., we anticipate context-induced conditionally independent parts represented in orthogonal subspaces. We call this the Factored World Hypothesis (FWH).

## A.5. Joint-to-factored map: reduced states and relative entropy

In general, valid predictive vectors are not constrained to the product-state submanifold in the joint representation. Many different off-manifold predictive vectors would map to the same point in the factored representation—such that the joint-to-factored map is non-invertible in general. Accordingly, *while the factored representation allows for a much lower-dimensional representation, it does not naturally accommodate a representation of correlated parts* and so would correspond to a lossy representation if the parts in fact remain correlated after conditioning on the token history.

### A.5.1. Marginalizing over subsystems to obtain reduced states

Given the joint predictive vector $\langle\boldsymbol{\eta}|$ and dynamics of a multipartite system, how do we take something like a "partial trace" over some subsystem(s) to obtain the reduced predictive state of the remaining subsystem(s)?

---

[11]It is helpful here to use bras $\langle\cdot|$ and kets $|\cdot\rangle$ to denote row vectors and column vectors, respectively. This allows for easy tracking of object type, as $|\cdot\rangle\langle\cdot|$ is a matrix and $\langle\cdot|\cdot\rangle$ is a scalar.

We achieve this via the $|\mathbf{1}_n\rangle$ vector(s) such that (i) $T_n|\mathbf{1}_n\rangle = |\mathbf{1}_n\rangle$, (ii) $|\mathbf{1}\rangle = \bigotimes_{m=1}^{N}|\mathbf{1}_m\rangle$, (iii) $T|\mathbf{1}\rangle = |\mathbf{1}\rangle$, and (iv) $\langle\boldsymbol{\eta}|\mathbf{1}\rangle = 1$. This allows us to "trace out" the $n^{\text{th}}$ subsystem via $\mathrm{tr}_n(\langle\boldsymbol{\eta}|) = \langle\boldsymbol{\eta}|\left(\otimes_{m=1}^{n-1}I_m\right)\otimes|\mathbf{1}_n\rangle\otimes\left(\otimes_{m=n+1}^{N}I_m\right)$.

Note that this contraction is indeed functionally equivalent to the partial trace of a density matrix for quantum systems, when our systems are quantum. The procedure introduced thus generalizes the notion of partial trace to marginalize over subsystems in generalized probabilistic theories (classical, quantum, or otherwise).

In the case of classical belief states, this is useful for finding the closest product state to an arbitrary joint state, as measured by the relative entropy (a.k.a., Kullback–Leibler divergence):

$$\mathrm{argmin}_{\boldsymbol{\eta}\in\boldsymbol{\mathcal{S}}_\otimes}\mathrm{D}\big[\boldsymbol{\eta}^{(x_{1:\ell})}\|\boldsymbol{\eta}\big] = \bigotimes_{n=1}^{N}\boldsymbol{\eta}_n^{(x_{1:\ell})}\,, \tag{14}$$

where $\boldsymbol{\mathcal{S}}_\otimes = \{\otimes_{n=1}^{N}\langle\boldsymbol{\eta}_n| : \langle\boldsymbol{\eta}_n|\in\boldsymbol{\mathcal{S}}_n\}$ is the set of product states. The answer is the tensor product of all of the reduced states, where the reduced state of the $n^{\text{th}}$ subsystem traces over all subsystems *except* for the $n^{\text{th}}$ subsystem:

$$\langle\boldsymbol{\eta}_n^{(x_{1:\ell})}| := \langle\boldsymbol{\eta}^{(x_{1:\ell})}|\left(\otimes_{m=1}^{n-1}|\mathbf{1}_m\rangle\right)\otimes I_n\otimes\left(\otimes_{m=n+1}^{N}|\mathbf{1}_m\rangle\right)\,. \tag{15}$$

It is important to note that the closest product state as measured by relative entropy is generally *not* the same as the closest product state as measured by Euclidean distance.

With the reduced marginals in hand, we can now quantify the **total correlation** $\mathrm{C}[\boldsymbol{\eta}]$ among all $N$ parts via

$$\mathrm{C}[\boldsymbol{\eta}] := \mathrm{D}\Big[\boldsymbol{\eta}\Big\|\bigotimes_{n=1}^{N}\boldsymbol{\eta}_n\Big]\,, \tag{16}$$

where $\boldsymbol{\eta}_n$ is the reduced state of the $n^{\text{th}}$ subsystem. Properly adapted, Eq. (16) is consistent with both classical and quantum information theory, and can be applied more generally to mixed representations.

A.5.2. JOINT-TO-FACTORED MAP

The linear map from joint to factored representations can now be written explicitly via combining partial trace with the linear embedding maps from the previous section:

$$\langle\boldsymbol{\eta}_{\mathrm{FWH}}^{(x_{1:\ell})}| = \langle\boldsymbol{\eta}^{(x_{1:\ell})}|\sum_{n=1}^{N}\bigg[\left(\otimes_{m=1}^{n-1}|\mathbf{1}_m\rangle\right)\otimes I_n\otimes\left(\otimes_{m=n+1}^{N}|\mathbf{1}_m\rangle\right)\bigg]\iota_n\,. \tag{17}$$

# B. Minimal explicit example of joint vs. factored representations

Here, we work through the mathematics and geometry of a minimal example that compares the joint and factored representations.[12]

A nontrivial example requires at least two factors, each with at least a two-dimensional latent space. Accordingly, for the independent case, we take our minimal nontrivial example to be a process with two factors that are each non-Markovian 2-state HMMs with a binary alphabet.

We take each of these constituent HMMs to be a Simple Nonunifilar Source (SNS) (Travers & Crutchfield, 2011), with labeled transition matrices

$$T_n^{(0)} = \begin{bmatrix} 0 & 0 \\ q_n & 0 \end{bmatrix}\,, \quad T_n^{(1)} = \begin{bmatrix} 1-p_n & p_n \\ 0 & 1-q_n \end{bmatrix} \tag{18}$$

for $n\in\{1,2\}$, with $p_n, q_n\in(0,1)$. Each of these factors can be thought of as producing 'subtokens' from the binary alphabet $\mathcal{Z}_1 = \mathcal{Z}_2 = \{0,1\}$. The net row-stochastic transition matrix for each factor $T_n = T_n^{(0)} + T_n^{(1)} = \begin{bmatrix} 1-p_n & p_n \\ q_n & 1-q_n \end{bmatrix}$

---

[12]One possibly misleading part of this minimal example is that the number of dimensions in the joint and factored representations don't seem too different. For more factors or with higher dimensions for each, the exponential discrepancy quickly becomes obvious.

has the stationary right eigenvector $\mathbf{1}_n = T_n \mathbf{1}_n = [1,1]^\top$ (which can be thought of as integrating latent probability) and stationary left eigenvector $\boldsymbol{\pi}_n = \boldsymbol{\pi}_n T_n = \frac{1}{p_n + q_n}[q_n, p_n]$ (which can be thought of as the stationary distribution over hidden states if marginalizing over subtokens).

Observed tokens $\mathcal{X} = \{A, B, C, D\}$ correspond to the possible combinations of these subtokens: $00 \mapsto A$, $01 \mapsto B$, $10 \mapsto C$, and $11 \mapsto D$. The minimal GHMM for this process with two independent factors has transition operators acting on the joint $\mathbb{R}^4$ space: $T^{(A)} = T_1^{(0)} \otimes T_2^{(0)}$, $T^{(B)} = T_1^{(0)} \otimes T_2^{(1)}$, $T^{(C)} = T_1^{(1)} \otimes T_2^{(0)}$, and $T^{(D)} = T_1^{(1)} \otimes T_2^{(1)}$.

The net row-stochastic transition matrix on the joint latent space $T = \sum_{x \in \{A,B,C,D\}} T^{(x)} = T_1 \otimes T_2$ has the stationary right eigenvector $\mathbf{1} = T\mathbf{1} = (T_1 \otimes T_2)\mathbf{1} = \mathbf{1}_1 \otimes \mathbf{1}_2 = [1,1,1,1]^\top$, (which again can be thought of as integrating latent probability) and stationary left eigenvector $\boldsymbol{\pi} = \boldsymbol{\pi} T = \boldsymbol{\pi}(T_1 \otimes T_2) = \boldsymbol{\pi}_1 \otimes \boldsymbol{\pi}_2 = \frac{1}{(p_1+q_1)(p_2+q_2)}[q_1 q_2, q_1 p_2, p_1 q_2, p_1 p_2]$ (which can be thought of as the stationary distribution over joint hidden states if marginalizing over token observations). We take the initial predictive vector to be the stationary state $\boldsymbol{\eta}^{(\varnothing)} = \boldsymbol{\pi} = \boldsymbol{\pi}_1 \otimes \boldsymbol{\pi}_2 = \boldsymbol{\eta}_1^{(\varnothing)} \otimes \boldsymbol{\eta}_2^{(\varnothing)}$.

We can now compare two candidate representations:

(i) The **joint representation**, where each predictive state $\boldsymbol{\eta}^{(x_{1:\ell})} = \frac{\boldsymbol{\eta}^{(\varnothing)} T^{(x_{1:\ell})}}{\boldsymbol{\eta}^{(\varnothing)} T^{(x_{1:\ell})} \mathbf{1}}$ lives on the curved 2d manifold of product states $\{[p, 1-p] \otimes [p', 1-p'] : p, p' \in [0,1]\}$ within the 3-dimensional 3-simplex[13] $\Delta^3$ embedded in the joint space $\mathbb{R}^4$. The collection of context-induced predictive states $\{\boldsymbol{\eta}^{(w)} : w \in \mathcal{X}^*\}$ *spans* $\mathbb{R}^4$ but only *varies* in three directions since differences in predictive states are all orthogonal to $\mathbf{1}$—since $(\boldsymbol{\eta}^{(w)} - \boldsymbol{\eta}^{(w')})\mathbf{1} = \boldsymbol{\eta}^{(w)}\mathbf{1} - \boldsymbol{\eta}^{(w')}\mathbf{1} = 1 - 1 = 0$.

(ii) The **factored representation**, where a linear transformation of the predictive state preserves all information (via a one-to-one mapping, *as long as the predictive state lives on the product-state manifold*) while producing a lower-dimensional representation. The factored representation allocates an orthogonal subspace to the latent probability simplex of each factor. In this case, since each factor has a two-dimensional latent space, the factored representation accommodates each 1-simplex with its own dimension. Since $[1, -1]^\top$ is orthogonal to $\mathbf{1}_n = [1, 1]^\top$, the degree of freedom for each factor is the scalar coefficient for $|b_{2,n}\rangle = [1, -1]^\top$. The factored representation takes the form

$$\boldsymbol{\eta}_{\text{FWH}}^{(w)} = \iota_1(\boldsymbol{\eta}_1^{(w)}) + \iota_2(\boldsymbol{\eta}_2^{(w)}) \tag{19}$$

where $\iota_n : \underbrace{\mathbb{R}^{d_n}}_{\mathbb{R}^2} \to \underbrace{\mathbb{R}^{\sum_{n'=1}^{N}(d_{n'}-1)}}_{\mathbb{R}^2}$ linearly embeds each local predictive vector into a shared 2-dimensional embedding space. In particular, the embedding map takes the form $\iota_n = |b_{2,n}\rangle\langle\delta_{1+n}| = \frac{1}{2}\begin{bmatrix} 1 \\ -1 \end{bmatrix}\langle\delta_{1+n}|$, where $\langle\delta_k|$ is a row vector which is one-hot at its $k^{\text{th}}$ element. Altogether, this yields the factored representation

$$\boldsymbol{\eta}_{\text{FWH}}^{(w)} = \frac{1}{2}\boldsymbol{\eta}_1^{(w)}\begin{bmatrix} 1 \\ -1 \end{bmatrix}\begin{bmatrix} 1 & 0 \end{bmatrix} + \frac{1}{2}\boldsymbol{\eta}_2^{(w)}\begin{bmatrix} 1 \\ -1 \end{bmatrix}\begin{bmatrix} 0 & 1 \end{bmatrix}. \tag{20}$$

Note that all points in the factored representation live in the square (ranging from $-\frac{1}{2}$ to $\frac{1}{2}$ on each side), which is just a global shift of the Cartesian product of the constituent 1-simplices $\Delta^1 \times \Delta^1$.

Fig. 5 shows these two candidate ground-truth representations side by side.

### B.1. Dependent chain

A minimal dependent case with nontrivial candidate geometries can be obtained via a simple adaptation of the previous independent example. The first factor remains the 2-state SNS HMM. However, the second factor now depends on the first.

One way to realize this is by making the second factor an *input-dependent* HMM with conditioned transition matrices $T_2^{(z^{(2)}|z^{(1)})}$, where the transition parameters $p_2$ and $q_2$ now depend on the output $z^{(1)}$ of the first factor. Observed tokens $\mathcal{X} = \{A, B, C, D\}$ again correspond to the possible combinations of these subtokens: $00 \mapsto A$, $01 \mapsto B$, $10 \mapsto C$, and $11 \mapsto D$. However, the minimal GHMM for this process with one independent and one dependent factor has transition

---

[13]Recall that the $n$-simplex, $\Delta^n = \{(v_j)_{j=1}^{n+1} \in \mathbb{R}^{n+1} : \sum_{j=1}^{n+1} v_j = 1 \text{ and } v_j \geq 0\}$, corresponds to the set of all probability distributions over $n + 1$ elements. Above, the 3-simplex is the tetrahedron containing all probability distributions over the four joint hidden states of the minimal GHMM. Similarly, the 1-simplex is a line segment serving to contain all probability distributions over the two hidden states of one of the SNS factors, if we marginalize over the other factor.

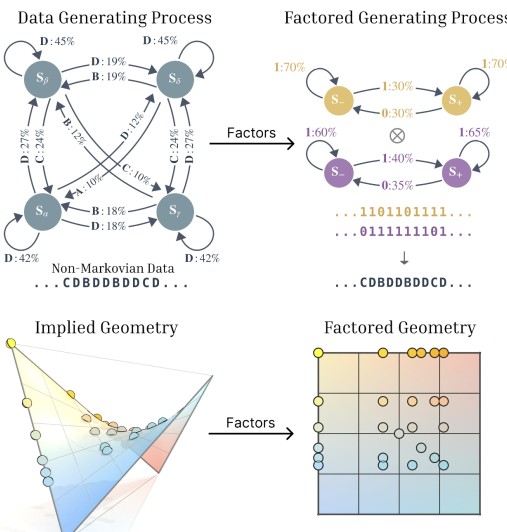

*Figure 5.* A minimal example (two SNS factors) to compare two candidate representations (joint vs. factored). The joint representation (bottom left) is a curved 2d submanifold of a 3-simplex, and varies in three dimensions. The factored representation (bottom right) contains belief updates over each factor in an orthogonal subspace, and so only varies in two dimensions.

operators acting on the joint $\mathbb{R}^4$ space: $T^{(A)} = T_1^{(0)} \otimes T_2^{(0|0)}$, $T^{(B)} = T_1^{(0)} \otimes T_2^{(1|0)}$, $T^{(C)} = T_1^{(1)} \otimes T_2^{(0|1)}$, and $T^{(D)} = T_1^{(1)} \otimes T_2^{(1|1)}$.

It's now useful to define $T_2^{|z^{(1)}} = \sum_{z^{(2)} \in \mathcal{Z}_2} T_2^{(z^{(2)}|z^{(1)})} = T_2^{(0|z^{(1)})} + T_2^{(1|z^{(1)})}$, which is row-stochastic for each $z^{(1)} \in \mathcal{Z}_1$.

The net row-stochastic transition matrix on the joint latent space $T = \sum_{x \in \{A,B,C,D\}} T^{(x)} = T_1^{(0)} \otimes T_2^{|0} + T_1^{(1)} \otimes T_2^{|1}$ has the stationary right eigenvector $\mathbf{1} = T\mathbf{1} = (T_1^{(0)} \otimes T_2^{|0} + T_1^{(1)} \otimes T_2^{|1})\mathbf{1} = \mathbf{1}_1 \otimes \mathbf{1}_2 = [1,1,1,1]^\top$, (which once again can be thought of as integrating latent probability).

Although the stationary right eigenvector $\mathbf{1}$ can be decomposed as a product state for a compositional GHMM[14], this is not generally true of the stationary *left* eigenvector $\boldsymbol{\pi}$. In this case, $\boldsymbol{\pi}$ is only a product state if $p_2/q_2$ is independent of $x^{(1)}$, which would then yield the same local stationary distribution $\boldsymbol{\pi}_2$ for both $T_2^{|0}$ and $T_2^{|1}$, and would imply $\boldsymbol{\pi} = \boldsymbol{\pi}_1 \otimes \boldsymbol{\pi}_2$. Else, if $p_2/q_2$ depends on $x^{(1)}$, then the joint stationary distribution $\boldsymbol{\pi}$ will be *off of the product-state manifold*, corresponding to mutual information between the two subsystems.

In any case, if we're designing the stochastic process, we could choose for the initial joint distribution to be some non-stationary product state, like $\boldsymbol{\eta}^{(\varnothing)} = [1/2, 1/2]^{\otimes 2}$. Upon subsequent observations, the predictive state would then remain on the product-state manifold—so either the joint or more concise factored representation would suffice for the best possible predictions, as indicated in Fig. 6.

### B.2. Forecasting correlated factors requires a joint representation space

If you pick a predictive vector uniformly from the 3-simplex, you *will* obtain a predictive vector that is off of the product-state manifold. For our example of two SNS subsystems, this predictive vector corresponds to a joint distribution over latent-state pairs encoding nonzero mutual information between the two subsystems. In other words, the product states are a measure-zero set within the broader set of joint states. From this perspective, it is quite special for predictive vectors to be on the product-state manifold. Indeed, general update maps correlate the parts, even if they were initially uncorrelated.

---

[14]More specifically, the stationary right eigenvector $\mathbf{1}$ can be decomposed as a product state for a compositional GHMM when the *type* of factor is fixed for each $n$—e.g., always an HMM of a fixed size, or always operating on a quantum subsystem of fixed Hilbert-space dimension—such that $\mathbf{1}_n$ is independent of $x$.

Conditionally Independent Parts

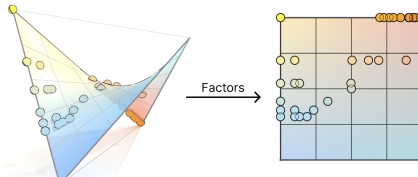

*Figure 6.* A chain of dependent factors still has a lossless factored representation, since there is a one-to-one linear map from the product-state manifold in the joint space to the Cartesian product of constituent simplices. In contrast to the independent case, the joint distribution over the two 1-simplices can no longer be obtained from the outer-product of the two marginal distributions. (Hence the lack of symmetry of points in the square compared with the independent example.)

So general joint states are relevant. How 'far' are these general joint states from product states? Euclidean distance in the 3-simplex may be a misleading metric. The total correlation may be more relevant. And, for the purpose of predictive loss, relative entropy of next-token predictions is yet more directly relevant.

We will work through these details for the case of two SNS subsystems, as an extension of the independent and conditionally independent example above.

For an arbitrary joint probability $\boldsymbol{\eta} \in \Delta^3$ in the 3-simplex, the closest product state—as measured by relative entropy—is the product of its two marginal distributions. In the language of GHMMs and Sec. A.5, this is the tensor product of the two reduced states. The mutual information between the latent states of the two subsystems is

$$I[\boldsymbol{\mathcal{S}}_1; \boldsymbol{\mathcal{S}}_2] = D[\boldsymbol{\eta} \| \boldsymbol{\eta}_1 \otimes \boldsymbol{\eta}_2] \,, \tag{21}$$

where $\boldsymbol{\eta}_1 = \boldsymbol{\eta}(I_1 \otimes \mathbf{1}_2)$ and $\boldsymbol{\eta}_2 = \boldsymbol{\eta}(\mathbf{1}_1 \otimes I_2)$.

By standard Euclidean distance, the closest product-state to any joint state is along a line segment normal (i.e., perpendicular) to the local plane of the product-state manifold. However, according to the minimal relative entropy, there is a fixed direction in the 3-simplex from any joint state to its closest product state—the product of its marginals—as shown in Fig. 7.[15]

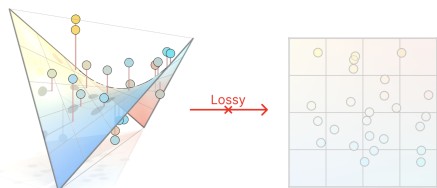

*Figure 7.* Infinitely many distinct joint states map to the same reduced state on the product-state manifold. These off-manifold joint states encode mutual information between the latent states of the subsystems.

## C. Processes used in the experiments

### C.1. Factors: Mess3 and Bloch Walk Definition

Here we define the Mess3 and Bloch Walk processes that serve as factors throughout the paper.

#### C.1.1. MESS3 PROCESS

The Mess3 process (Marzen & Crutchfield, 2017; Shai et al., 2024; Piotrowski et al., 2025) has three hidden states and three observable tokens $\mathcal{X} = \{0, 1, 2\}$. Since Mess3 is a 3-state HMM, its local predictive vectors live in a 2-simplex. This 2-simplex corresponds to the set of probability distributions over the three hidden states.

The Mess3 process is defined by two parameters, $\alpha$ and $x$, with dependent quantities $\beta = (1 - \alpha)/2$ and $y = 1 - 2x$.

---

[15]The direction from a joint state to its product of marginals in the 3-simplex is independent of the joint state and proportional to $[1, -1, -1, 1]^\top$. However, in higher dimensions, where there are more degrees of freedom, the direction depends on the joint state.

The labeled transition matrices are:

$$T^{(0)} = \begin{bmatrix} \alpha y & \beta x & \beta x \\ \alpha x & \beta y & \beta x \\ \alpha x & \beta x & \beta y \end{bmatrix} \tag{22}$$

$$T^{(1)} = \begin{bmatrix} \beta y & \alpha x & \beta x \\ \beta x & \alpha y & \beta x \\ \beta x & \alpha x & \beta y \end{bmatrix} \tag{23}$$

$$T^{(2)} = \begin{bmatrix} \beta y & \beta x & \alpha x \\ \beta x & \beta y & \alpha x \\ \beta x & \beta x & \alpha y \end{bmatrix} . \tag{24}$$

The net transition matrix

$$T = T^{(0)} + T^{(1)} + T^{(2)} = (\alpha + 2\beta) \begin{bmatrix} 1 - 2x & x & x \\ x & 1 - 2x & x \\ x & x & 1 - 2x \end{bmatrix} \tag{25}$$

has a uniform stationary distribution $\boldsymbol{\pi} = \boldsymbol{\pi} T = \frac{1}{3} \begin{bmatrix} 1 & 1 & 1 \end{bmatrix}$, and a stationary right eigenstate of $\mathbf{1} = T\mathbf{1} = \begin{bmatrix} 1 & 1 & 1 \end{bmatrix}^\top$.

### C.1.2. BLOCH WALK PROCESS

Bloch Walk (Riechers et al., 2025) has no finite HMM representation but can be simulated via interactions with a single qubit of quantum memory. It thus has a simple GHMM and its predictive vectors all lie on the $x$-$z$ slice of the Bloch ball.

In general, valid states of a qubit can be described by the extended Bloch vector $\vec{b} = [1, b_x, b_y, b_z]$. Interactions with a qubit induce linear transformations on this Bloch vector—and projective transformations, once the state is properly renormalized.

The Bloch Walk process is parametrized by $\alpha > 0$ and $\beta \in \mathbb{R}$. Let $\gamma = 1/(2\sqrt{\alpha^2 + \beta^2})$.

The minimal three-dimensional GHMM for this Bloch Walk process is obtained by projecting out the non-utilized $y$-component of the Bloch ball:

$$T^{(0)} = \begin{bmatrix} 1/4 & 0 & 2\alpha\beta\gamma^2 \\ 0 & (\alpha^2 - \beta^2)\gamma^2 & 0 \\ 2\alpha\beta\gamma^2 & 0 & 1/4 \end{bmatrix} \tag{26}$$

$$T^{(1)} = \begin{bmatrix} 1/4 & 0 & -2\alpha\beta\gamma^2 \\ 0 & (\alpha^2 - \beta^2)\gamma^2 & 0 \\ -2\alpha\beta\gamma^2 & 0 & 1/4 \end{bmatrix} \tag{27}$$

$$T^{(2)} = \begin{bmatrix} 1/4 & 2\alpha\beta\gamma^2 & 0 \\ 2\alpha\beta\gamma^2 & 1/4 & 0 \\ 0 & 0 & (\alpha^2 - \beta^2)\gamma^2 \end{bmatrix} , \text{ and} \tag{28}$$

$$T^{(3)} = \begin{bmatrix} 1/4 & -2\alpha\beta\gamma^2 & 0 \\ -2\alpha\beta\gamma^2 & 1/4 & 0 \\ 0 & 0 & (\alpha^2 - \beta^2)\gamma^2 \end{bmatrix} , \tag{29}$$

which can be interpreted as acting from the right on the coefficients $[c, b_x, b_z]$ of the ordered operator basis $(I/2, \sigma_x/2, \sigma_z/2)$, where $\sigma_x$ and $\sigma_z$ are two of the familiar Pauli matrices from quantum mechanics. See Riechers et al. (2025) for further details of this process.

The net transition operator

$$T = \sum_{x \in \mathcal{X}} T^{(x)} = \begin{bmatrix} 1 & 0 & 0 \\ 0 & 1/2 + 2(\alpha^2 - \beta^2)\gamma^2 & 0 \\ 0 & 0 & 1/2 + 2(\alpha^2 - \beta^2)\gamma^2 \end{bmatrix} \tag{30}$$

has the stationary left eigenvector $\boldsymbol{\pi} = \boldsymbol{\pi} T = \begin{bmatrix} 1 & 0 & 0 \end{bmatrix}$ and stationary right eigenvector $\mathbf{1} = T\mathbf{1} = \begin{bmatrix} 1 & 0 & 0 \end{bmatrix}^\top$.

C.1.3. JOINT PROCESSES COMBINE THESE SUBSYSTEMS

Examples in the main text used three Mess3 factors and two Bloch Walk factors. The joint representation spans $3^5 = 243$ dimensions, (but varies in only 242 dimensions, since it is restricted to the 242-simplex of this 243-dimensional space). The factored representation spans $5 \times 2 = 10$ dimensions, and can be seen as the Cartesian product of three 2-simplices $\Delta^2$ and two 2-balls $\circ^2$, with a collective space $\Delta^2 \times \Delta^2 \times \Delta^2 \times \circ^2 \times \circ^2$.

## C.2. Independent

Here we describe the independent factors experiments that test whether transformers learn factored representations when trained on sequences generated by a product of independent stochastic processes. In this setting, we compose five factors—three Mess3 processes operating on 2-simplices and two Bloch Walk processes operating on Bloch spheres—where each factor evolves according to its own transition dynamics and emits tokens independently.

Each factor $n$ has a 3-dimensional latent space and emits subtokens $z^{(n)} \in \mathcal{Z}_n$ according to its own emission distribution. For the Mess3 factors ($n \in \{1, 2, 3\}$), the subtokens are $\mathcal{Z}_n = \{0, 1, 2\}$; for the Bloch Walk factors ($n \in \{4, 5\}$) the subtokens are, $\mathcal{Z}_n = \{0, 1, 2, 3\}$. subtokens combine into a single observed token by mapping the Cartesian product to integers: $x \in \mathcal{X} = \{0, 1, \ldots, 431\}$, giving a vocabulary of size $|\mathcal{X}| = 3^3 \times 4^2 = 432$. As an explicit construction of the tokens from subtokens, we can take $x_\ell = \left( \sum_{n=1}^{3} z_\ell^{(n)} 3^{n-1} \right) + \left( \sum_{n=4}^{5} z_\ell^{(n)} 3^3 4^{n-4} \right) \in \mathcal{X} = \{0, 1, \ldots, 431\}$.

Because the factors are independent, the joint transition operator takes the tensor product form

$$T^{(x)} = \bigotimes_{n=1}^{N} T_n^{(z^{(n)})}, \tag{31}$$

and predictive vectors remain product states throughout belief updating. The joint latent space is the tensor product of the five 3-dimensional factor spaces, giving dimension $3^5 = 243$. Since predictive vectors are normalized, they vary in $243 - 1 = 242$ dimensions. In contrast, the factored representation tracks each factor's predictive vector separately in orthogonal subspaces; since each factor's predictive vector is also normalized, each varies in $3 - 1 = 2$ dimensions, giving $5 \times 2 = 10$ total dimensions.

*Table 1.* Process configurations for the independent factor experiments.

| Factors | Process | Parameters |
|---|---|---|
| $F_0, F_1, F_2$ | Mess3 | $\alpha = 0.6, \ x = 0.15$ |
| $F_3, F_4$ | Bloch Walk | $\alpha = 1.0, \ \beta = 3.0$ |

Table 1 summarizes the process parameters. Please notice and forgive the abuse of notation for both $x$ and $\alpha$.

## C.3. Conditionally independent

Because the condition for a lossless projection to the product state is *conditional* independence, we expect that models will find the factored representation when the factors depend on each other, provided their joint evolution can be described by equation (4). To investigate this we investigate a feed-forward dependency chain where each factor's dynamics in the latent space depend on preceding factors' outputs, but tensor product structure is preserved: $T^{(x)} = \bigotimes_{n=1}^{N} T_n^{(z^{(n)} | z^{(1:n-1)})}$.

Following section A.4, there must exist a lossless linear map from the product-state sub-manifold to a factored tensor product representation where the local predictive vectors for each subspace live in orthogonal subspaces:

$$\boldsymbol{\eta}_{\text{FWH}}^{(x_{1:\ell})} = \bigoplus_{n=1}^{N} \boldsymbol{\eta}_n^{(x_{1:\ell})} = \left( \tilde{\boldsymbol{\eta}}_1^{(x_{1:\ell})}, \tilde{\boldsymbol{\eta}}_2^{(x_{1:\ell})}, \ldots, \tilde{\boldsymbol{\eta}}_N^{(x_{1:\ell})} \right). \tag{32}$$

We train a transformer on data generated from a chain of three Mess3 processes and two Bloch Walk processes, similar to our earlier experiments—but now each of the subprocesses depends on [all of] the factors above it in the chain. As training data, we use tokens $x$ with a one-to-one correspondence to the Cartesian product of subtokens.

*Table 2.* Process configurations for the conditionally independent factor experiments. Left: parameters for each process variant. Right: variant selection for each factor, where the control variable $C$ is the subtoken emitted by the preceding factor.

| ID | Process | Parameters |
|----|---------|------------|
| M1 | Mess3 | $x = 0.15, \; \alpha = 0.60$ |
| M2 | Mess3 | $x = 0.11, \; \alpha = 0.79$ |
| M3 | Mess3 | $x = 0.50, \; \alpha = 0.60$ |
| B1 | Bloch Walk | $\alpha = 1.0, \; \beta = 2.00$ |
| B2 | Bloch Walk | $\alpha = 1.0, \; \beta = 2.50$ |
| B3 | Bloch Walk | $\alpha = 1.0, \; \beta = 3.00$ |
| B4 | Bloch Walk | $\alpha = 1.0, \; \beta = 3.50$ |

| Factor | Control | Variants |
|--------|---------|----------|
| $F_0$ | — | M1 |
| $F_1$ | $C \in \{0, 1, 2\}$ | M1, M2, M3 |
| $F_2$ | $C \in \{0, 1, 2\}$ | M1, M2, M3 |
| $F_3$ | $C \in \{0, 1, 2\}$ | B1, B2, B3 |
| $F_4$ | $C \in \{0, 1, 2, 3\}$ | B1, B2, B3, B4 |

Table 2 shows the parameters for each process variant and the mapping from control subtokens to variants. The variation in $\beta$ for Bloch Walk and in both $x$ and $\alpha$ for Mess3 ensures that different variants induce meaningfully different dynamics, testing whether the model can track factor-specific predictive vectors that evolve under context-dependent transition rules.

## C.4. Almost independent example: Noised factors

Here we describe the noised factors experiments that test whether transformers maintain factored representations even when the data is not fully consistent with a factorized form. In this setting, the optimal predictive distribution lies off the product-state manifold in the joint belief space, meaning that cross-entropy loss can in principle be decreased by leveraging the higher-dimensional joint space rather than maintaining strict factorization.

This is an example of an indecomposable process, which has the general form: $T^{(x)} = \varepsilon T_{\text{int}} + (1 - \varepsilon) \bigotimes_{n=1}^{N} T_n^{(x)}$.

We use the same five independent factors as described above, but now each emitted token $x \in \mathcal{X} = \{0, 1, \ldots, 431\}$ passes through a memoryless noisy channel with probability $\varepsilon$ of being replaced uniformly by any other token.

To describe the token-labeled transition matrices explicitly, it is useful to define $\mathcal{Z} = \bigtimes_{n=1}^{N} \mathcal{Z}_n$. The transition operators on the joint latent space then take the form

$$T^{(x)} = (1 - \varepsilon)\Big(\bigotimes_{n=1}^{N} T_n^{(z^{(n)})}\Big) + \frac{\varepsilon}{|\mathcal{X}| - 1} \sum_{z'^{(1:N)} \in \mathcal{Z} \setminus \{z^{(1:N)}\}} \bigotimes_{n=1}^{N} T_n^{(z'^{(n)})} . \tag{33}$$

We train models across noise levels $\varepsilon \in \{0, 0.001, 0.01, 0.05, 0.1, 0.2, 0.3, 0.5\}$, spanning from the fully factored regime ($\varepsilon = 0$) to a highly corrupted setting where half of all tokens are randomized.

For low $\varepsilon$, we find that the model first learns a factored representation, which does a relatively good job at prediction with relatively few dimensions. To get below a critical loss threshold (which is a function of $\varepsilon$), the model then slowly increases the number of effective dimensions of its residual stream that it uses.

What is going on? For low $\varepsilon$, this noised process is *almost* product preserving, and so the predictive vectors remain close to the product-state manifold. The projection down to the factored representation is not too lossy. But, as the model receives pressure to lower loss, or if $\varepsilon$ is large enough such that the projection is quite lossy, the model is pressured to learn the higher-dimensional representation that can losslessly encode off-manifold predictive vectors.

## C.5. The loss gap between the full and the joint representation

In our experiments we introduce noise such that, with probability epsilon, each token is replaced by a token uniformly sampled from the alphabet. Therefore, we can think of there being two regimes, one where $\varepsilon < .5$, one where $.5 \le \varepsilon$. For $\varepsilon > .5$, the process becomes increasingly dominated by un-learnable memory-less noise. At the extreme case, $\varepsilon = 1$, the process becomes pure noise and the minimal generator becomes a one state HMM that uniformly emits over the joint vocabulary. In our experiments we consider the first regime where $0 < \varepsilon < .5$. Because we are corrupting our data using noise, one might wonder how it is possible for one predictive model to model it better than the other. Isn't any of this kind of noise, by definition, un-learnable?

The key subtlety is that the observer does not know which tokens were corrupted. After observing a token, the posterior

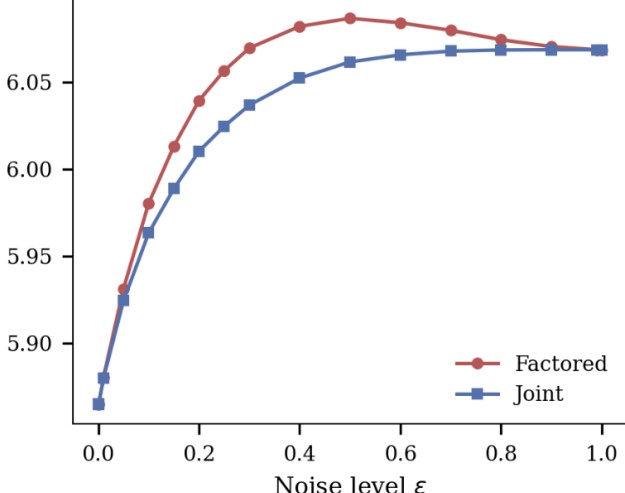

*Figure 8.* Average cross entropy loss from using both the joint and the factored representations to to next-token prediction noised data for different values of $\varepsilon$. At $\varepsilon = 0$, there is no corruption of the observed token, so the joint and factored processes are equally predictive. When $\varepsilon = 1$ all tokens are simply the result of noise, so there is no structure to predict and both models do equally well. **We test the regime where $0 < \varepsilon < .5$, to give the model an increasing loss penalty for choosing the factored representation rather than the joint**. Once $\varepsilon$ is larger than .5, the model has ever decreasing incentive to use an internal representation that relates to the structured generator at all as uniform noise begins to dominate generation.

belief is a mixture of the case that the observed token was a corruption and that it was faithful. With weight $(1 - \varepsilon)$ the token was faithful and each factor updates independently. With weight $\varepsilon$ it was noise and uninformative. Because this uncertainty applies jointly across all factors, the Bayesian updates entangle beliefs about different factors. This pushes predictive vectors off the product-state manifold in a way the factored representation cannot capture but the joint can. Increasing epsilon strengthens these cross-factor correlations, creating increasing pressure on the transformer to represent joint beliefs faithfully. See figure 8 for a plot of the loss penalty that a model has to pay for choosing the factored representation at different values of $\varepsilon$.

## D. Extended Experimental Results

### D.1. Details of Independent Experiments

In the main text we show that, over training, the model learns a structure with dimensionality and geometry that is consistent with the factored representation. In Figure 9, we show additionally that the formation of this structure during training is associated with a reduction in loss. See the figure caption for additional details.

### D.2. Conditionally independent factors

Figure 10 reveals that neural networks trained via next-token prediction on the resulting sequences automatically learn to factor their representation into parts, where uncertainty over the latent states of each part is encoded geometrically.

### D.3. Experiments with large capacity transformers

We show in panels 11(i) and 11(ii) that our factoring results hold even with longer context. Additionally, we observe in panel 11(iii) that even with $d_{\text{model}}$ far greater than the size necessary to accommodate the joint representation, the model still prefers the factored representation.

### D.4. Experiments with RNNs and LSTMs

To investigate whether factored representations are specific to the Transformer architecture or represent a general model preference, we also trained vanilla RNNs and LSTMs on the same five-factor independent process used in section 3.

In figure 12(i), we see that at $d_{\text{model}} = 32$, vanilla RNNs closely replicate the Transformer results. Linear regression can

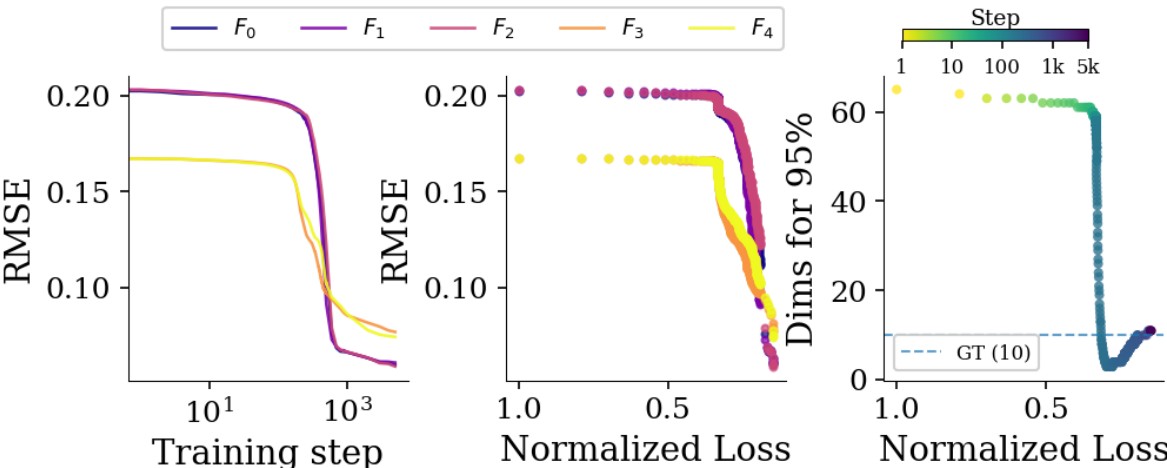

*Figure 9.* Early training dynamics. RMSE to the 5 independent factors, the number of dimensions needed to reach 95% CEV, and loss all decrease monotonically over training. (left) RMSE to each factor decreases monotonically over training (center) **RMSE to the predicted ground truth geometry is correlated with loss: as the regression improves, the loss consistently decreases.** (right) The model is able to consistently reduce loss by dropping dimensions until about 1000 steps. After this, it continues to reduce loss by converging to a dimensionality consistent with the factored representation. Further training results in a long slow decrease of loss and RMSE towards 0 with minimal change in effective dimensionality.

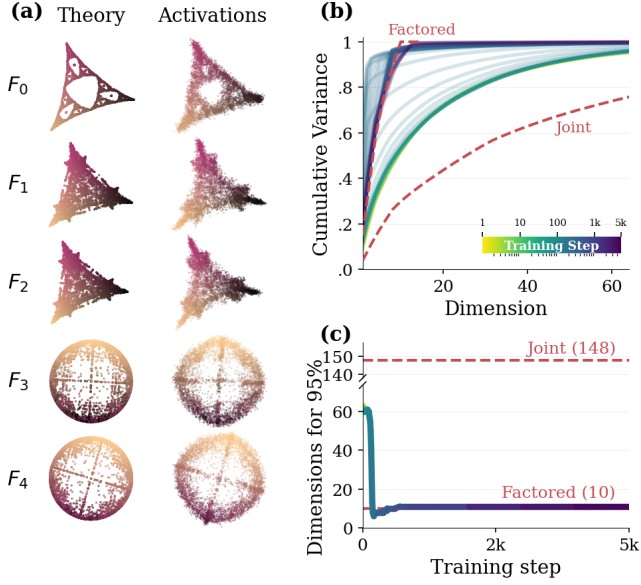

*Figure 10.* **Transformers learn to factor the world into parts, even when the parts form a chain of dependencies.** We train a transformer on sequences generated by five conditionally-independent factors; the model sees only tokens corresponding to the Cartesian product of factor outputs. (a) Left: ground-truth belief geometry for each factor. Right: linear regression from activations recovers each geometry with high fidelity. (b) Cumulative explained variance (CEV) over training. Dashed lines show predictions for factored (10D) versus joint (242D) representations; the model converges to the factored prediction. (c) Effective dimensionality (dimensions for 95% variance) collapses to 10 during training, far below the joint requirement.

recover representations of all 5 factors, and 10 components explain approximately 95% of the variance. As we increase the size to 64, we see in figure 12(ii) that the CEV curve smooths out, and the effective dimensionality settles around 16 dimensions. This is still far below the 136 components that we would expect to explain 95% of the joint representation, but notably higher than the expected 10 for the factored representation. This smoothing happens despite the model lacking capacity for the full joint representation. For LSTMs, we observe in figure 12(iii) that this smoothing happens even at $d_{\text{model}} = 32$. While we can recover geometries for all 5 factors, recurrent architectures do not appear to exhibit as sharp a preference for dimensional advantage that Transformers do. However, we see in figure 12(iv) that even when the RNN's size

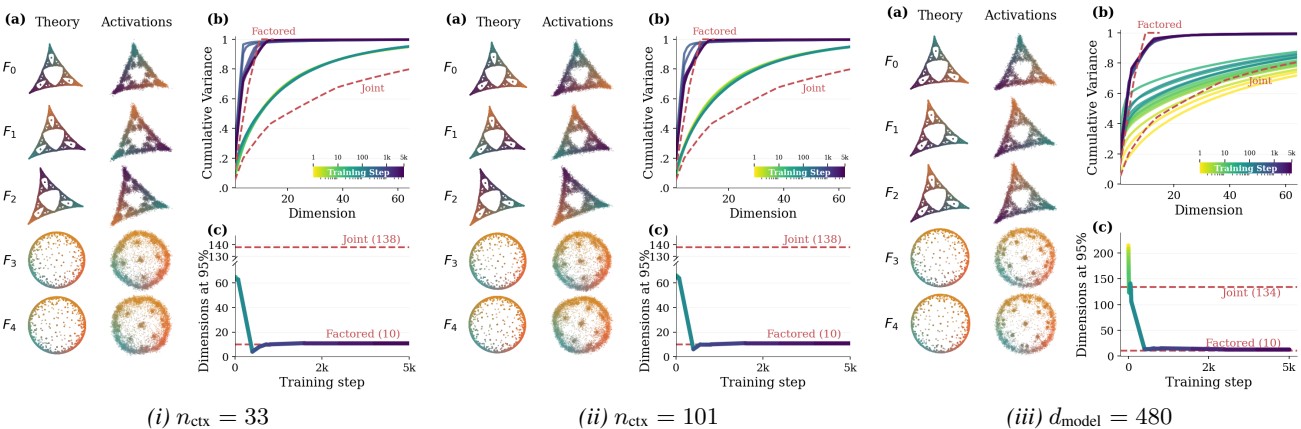

*Figure 11.* Control experiments: observations still hold for varying context windows and model dimension. Context length has no meaningful effect on our observations, as shown in panels i) and ii). Notably, panel iii) shows that **even when the model has enough dimensions to faithfully represent the entire joint representation, it finds and prefers the factored form.**

can fully accommodate the joint representation, it still does not appear to represent it. Rather, its CEV curve converges to one significantly closer to the factored representation. Whether the differences in architectures are due to training dynamics or inductive bias remains an open question.

### D.5. Additional noise experiments

In figure 13 we show the CEV-over-training curves for all reported noise levels from $\varepsilon = 0$ to $\varepsilon = 0.5$. We see that up to $\varepsilon = 0.05$, there is only a slight departure from the curve admitted by training on $\varepsilon = 0$. With noise $\varepsilon = 0.1$ and above, the curve gets softer and increasingly more dimensions are required to explain a large percentage of the cumulative variance.

### D.6. Factoring starts at the embedding matrix

Factorization starts all the way down at the token embedding, suggesting that the factored subspaces are used mechanistically throughout the layers.

We find that transformers learn a token embedding matrix that factors tokens into the relevant subspaces, ordered according to the learned subtoken decomposition. As seen in Fig. 14, there are around 12 non-negligible singular values of the learned token-embedding weight matrix. PCA of token embeddings reveals the emergence of crystalline structure over the course of training.

## E. Training details

We employ transformer models with learned positional embeddings and pre-norm architecture implemented through TransformerLens (Nanda & Bloom, 2022). Architecture sweeps were conducted over 1-4 layers, 1-4 heads, and $d_{\text{model}}$ sizes varying from 48 to 480. We use the convention that $d_{\text{ff}} = 4 \cdot d_{\text{model}}$ and $d_{\text{head}} = \frac{d_{\text{model}}}{n_{\text{heads}}}$ The main experimental results were generated on Transformers with 3 heads and 4 layers. For RNN and LSTM experiments, we used 2 layers and $d_{\text{model}}$ of 32, 64, and 256.

All models were trained on next-token prediction using cross-entropy loss, with a batch size of 25000. We reduce the batch size to 4096 for longer contexts (33 and 101) due to memory constraints. We use the Adam optimizer (Kingma & Ba, 2014), a learning rate of $5 \cdot 10^{-4}$, and no weight decay.

For the Transformer experiments we used a beginning-of-sequence (BOS) token at the start of each sequence. For RNN experiments we omitted this.

To regress to the factored belief states, we sample batches, and generate targets by concatenating the vectors associated with each factor's belief state. We then perform a forward pass on the batches, and do linear regression from model activations to those targets. We do not include the BOS-token as, for the Bloch Walk process, the initial state is not on the same manifold

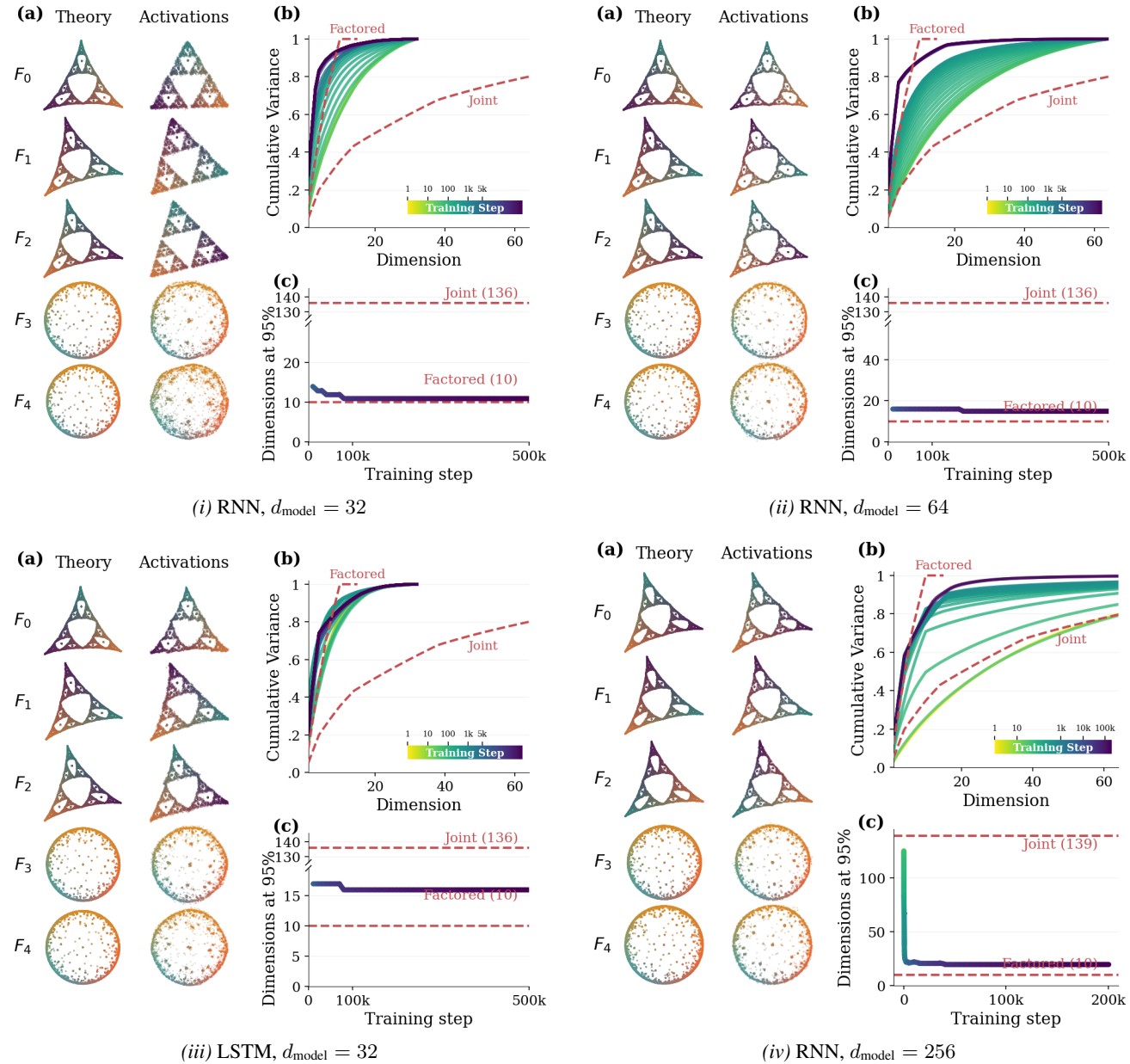

*Figure 12.* RNNs and LSTM experiments. The $d_{\mathrm{model}} = 32$ RNN matches $\sim 95\%$ CEV at 10 components. For $d_{\mathrm{model}} = 64$, the curve is smoother and uses more components to reach the same threshold. For an LSTM with $d_{\mathrm{model}} = 32$, we observe similar dynamics to an RNN of roughly twice the size, requiring more components to explain $95\%$. **Despite the smoother CEV curves, an RNN with capacity greater than the joint representation requires** (i.e. $d_{\mathrm{model}} = 256$) **still shows signs of factorization.**

that the belief states share.

Full training code and experiment configurations can be seen at: https://github.com/Astera-org/factored-reps

## F. Analysis: cumulative explained variance calculation

Given $M$ token sequences of context length $L$ in a network with $N_{\mathrm{layers}}$ layers and model dimension (e.g., residual-stream dimension) $d_{\mathrm{model}}$, we look at neural activations $\vec{a}_{x_{1:\ell}}$ of dimension $d_{\mathrm{model}}$ induced at the various context positions $\ell \in \{1, ..., L\}$ at a chosen layer. There are $M \cdot L$ of these activation vectors at each point in the residual stream, since we

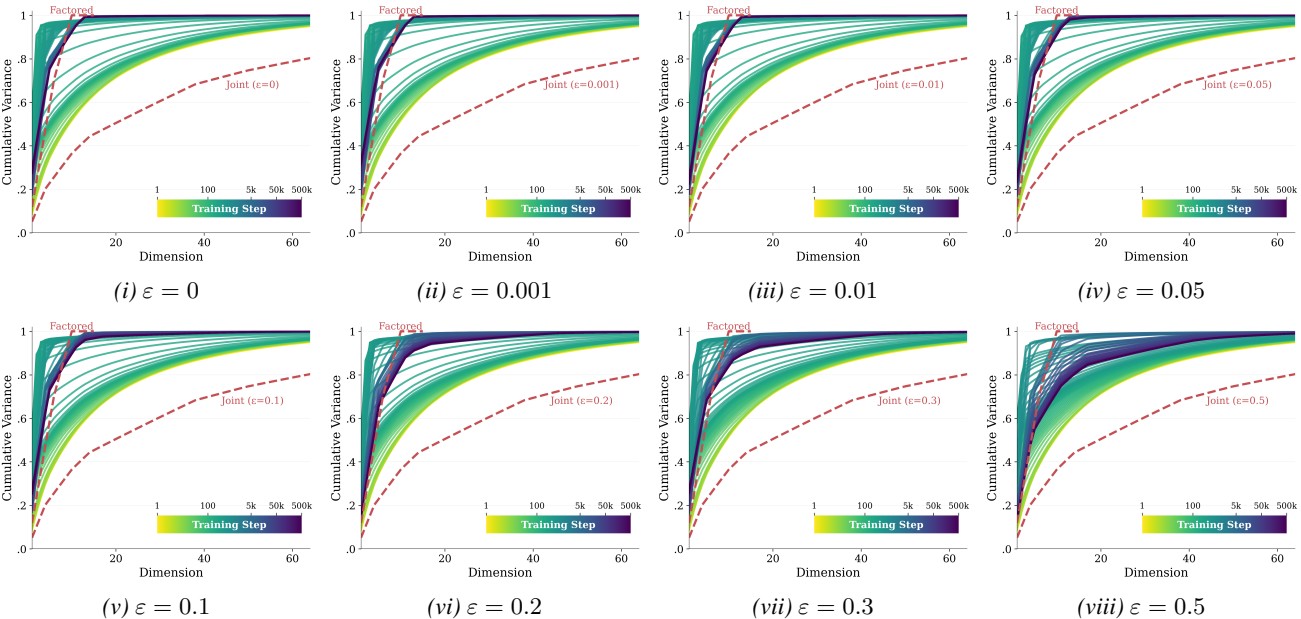

*Figure 13.* Cumulative Explained Variance (CEV) in residual stream activations across sequence positions for varying noise levels $\varepsilon$. As $\varepsilon$ increases, we see the model converge to higher and higher dimensional representations as the factored representation becomes increasingly lossy. **Despite this, the CEV curves reveal that the training dynamics reach the factored form early in training for each case.** Only for large $\varepsilon$, do we see the converged representation departs significantly from the factored form. And, even then, the converged CEV line looks much more like the ground truth for the factored representation than the joint one.

have $M$ sequences and each sequence induces $L$ vectors in the residual stream. Denoting these activations as row vectors $\langle a_i |$, they can be stacked into a data matrix $A = \sum_{i=1}^{ML} |\delta_i\rangle \langle a_i|$ where $|\delta_i\rangle$ is a one-hot column vector with the nonzero entry at the $i^{\text{th}}$ entry. Performing principal component analysis (PCA) on $A$ yields principal values, the eigenvalues $\lambda_j$ of the covariance matrix $A^\top A$ with $\lambda_{j+1} \geq \lambda_j$, as well as the corresponding orthonormal principal vectors. In order to interpret the eigenvalues as % of explained variance, we normalize by the sum of the eigenvalues $\Lambda$. From this, we define the *cumulative explained variance* (CEV):

$$\text{CEV}(k) = \frac{1}{\Lambda} \sum_{j=1}^{k} \lambda_j \ . \tag{34}$$

The number of principal components needed to attain a CEV threshold $p$ near unity quantifies the number of effective dimensions used by the model:

$$k_p^* \coloneqq \min_k \big( \{k : \text{CEV}(k) \geq p\} \big) \ . \tag{35}$$

In the main figures we report $k_{0.95}^*$ as "Dimensions for 95%".

Technically, activations span a $k_1^*$-dimensional subspace. Practically, however, there are typically many fewer dimensions that are actually used appreciably. For example, only a $k_{0.98}^*$-dimensional subspace is required to explain 98% of the variance of all activations.

Tracking the evolution of $k_p^*$ over training can be very informative about how many dimensions of the residual stream the model is using over the course of training. We find that $k_{0.95}^*$ neatly explains the discovery of low-dimensional subspaces for factored belief updating in our experiments.

We end by noting that for each point in the residual stream, we will have a different set of activations—and so we may build an $A$ matrix at each point (or, for that matter, any subset of points). In this work, unless noted otherwise, we look at $A$ generated from activations associated with the final transformer block, after MLP, and before the final layer norm. For example, in our 4 layer transformer implemented using TransformerLens, we grab activations using the hook `blocks.3.hook_resid_post`.

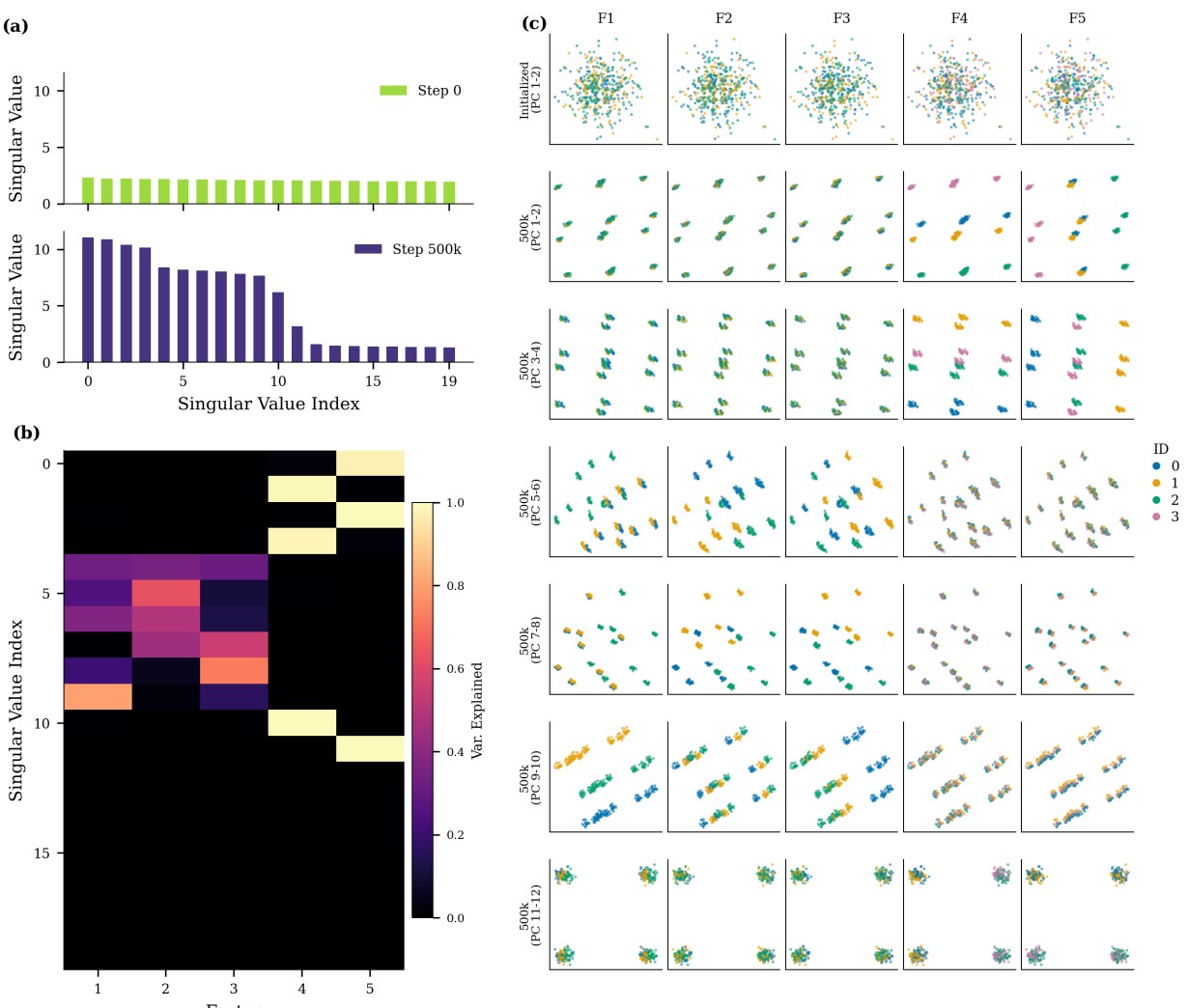

*Figure 14.* **The token-embedding weight matrix learns to partition the representation space according to the subtokens relevant to each factor.** (a) Visualizes the singular value spectrum of the embedding matrix at initialization (Step 0) versus after training (Step 500k). The trained embeddings exhibit a lower effective rank, consistent with factored structure. (b) Shows PCA projections of the token embeddings, with each point colored by subtoken identity (ID) for each factor (columns F1–F5). The top row shows randomly initialized embeddings (Step 0), which exhibit no structure. The remaining rows show trained embeddings (Step 500k) projected onto successive pairs of principal components (PC1–2 through PC11–12). We see that different PC pairs reveal different partitioning based on subtoken identities, indicating that the model learns a factored embedding with each factor encoded in a separate low-dimensional subspace. (c) Reveals the fraction of projection variance explained by each factor, for each singular vector ordered by decreasing singular value. The top 12 singular vectors show strong factor attribution (bright cells), while all subsequent rows are uniformly dark, indicating that factor-relevant structure is concentrated entirely within a low-dimensional subspace.

# G. Analysis: regression

Throughout our analyses, we use linear regression to probe the relationship between transformer activations and ground-truth predictive vectors. This appendix details our methodology.

## G.1. Least squares formulation

Given activations $A \in \mathbb{R}^{M \times d_{\text{model}}}$ from $M$ context realizations and corresponding ground-truth predictive vectors $Y \in \mathbb{R}^{M \times d}$, we fit a linear map:

$$\hat{Y} = AW + \mathbf{1}\boldsymbol{\beta}^\top \tag{36}$$

where $W \in \mathbb{R}^{d_{\text{model}} \times d}$ and $\boldsymbol{\beta} \in \mathbb{R}^d$. The objective is:

$$\mathcal{L}(W, \boldsymbol{\beta}) = \sum_{i=1}^{M} \|Y_i - (A_i W + \boldsymbol{\beta}^\top)\|^2 \, . \tag{37}$$

To solve this, we form the augmented design matrix $X = [\mathbf{1} \mid A] \in \mathbb{R}^{M \times (1 + d_{\text{model}})}$ and solve $\min_\theta \|X\theta - Y\|_{\text{F}}^2$ where $\theta = [\boldsymbol{\beta}; W]$.

### G.2. SVD-Based solution

We solve the system via singular value decomposition (SVD). Given $X = U\Sigma V^\top$, the solution is:

$$\hat{\theta} = V\Sigma^+ U^\top Y \tag{38}$$

where $\Sigma^+$ is the Moore–Penrose pseudoinverse of $\Sigma$.

For numerical stability, we apply singular value thresholding. Given a relative condition number threshold $\epsilon_{\text{rcond}}$, we set:

$$\sigma_i^{-1} \leftarrow \begin{cases} \sigma_i^{-1} & \text{if } \sigma_i > \epsilon_{\text{rcond}} \cdot \sigma_{\max} \\ 0 & \text{otherwise} \end{cases} \tag{39}$$

where $\sigma_{\max}$ is the largest singular value. To choose a threshold, we perform 10-fold cross validation, using multiple $\epsilon_{\text{rcond}}$ values (e.g., $\{10^{-15}, 10^{-10}, 10^{-8}, 10^{-6}, 10^{-4}, 10^{-2}\}$) and select the threshold minimizing mean reconstruction error over the held out test sets. We then take that threshold and apply it to the full dataset to get our final fit.

### G.3. Reconstruction quality metrics

We report reconstruction quality using root mean squared error (RMSE):

$$\text{RMSE} = \sqrt{\frac{1}{Md} \sum_{i=1}^{M} \sum_{j=1}^{d} (Y_{ij} - \hat{Y}_{ij})^2} \tag{40}$$

and coefficient of determination:

$$R^2 = 1 - \frac{\sum_{i,j} (Y_{ij} - \hat{Y}_{ij})^2}{\sum_{i,j} (Y_{ij} - \bar{Y}_j)^2} \tag{41}$$

where $\bar{Y}_j = \frac{1}{M} \sum_i Y_{ij}$ is the mean of target dimension $j$.

### G.4. Factored predictive vectors

When the generative process consists of $N$ independent factors, the ground-truth predictive vector decomposes as $Y = [Y^{(1)} | \cdots | Y^{(N)}]$ where $Y^{(n)} \in \mathbb{R}^{N \times d_n}$ is the predictive vector for factor $n$.

We use *joint regression*: a single regression from activations to the concatenated target $Y$, yielding coefficient matrix $W = [W^{(1)} | \cdots | W^{(N)}]$. This approach allows the model to use shared structure in activations for predicting multiple factors, and produces per-factor coefficient matrices $W^{(n)}$. We report both overall and per-factor RMSE.

## H. Analysis: factor subspace identification and orthogonality

Our theoretical framework predicts that if transformers learn factored representations, these representations should organize into orthogonal subspaces of activation space—one for each conditionally independent latent factor of the generative process. Testing this prediction requires (1) identifying candidate factor subspaces and (2) characterizing their geometric relationships. This section first presents two procedures for factor subspace identification—vary-one analysis and regression analysis—then describes two complementary analyses that leverage these identified subspaces to quantify orthogonality. The first orthogonality analysis computes the basis-invariant pairwise overlap between factor subspaces. The second tests whether effective dimensionalities are additive: specifically, whether the sum of effective dimensionalities of factor-specific representations equals that of their union, as orthogonality would require.

## H.1. Factor subspace identification

### H.1.1. VARY-ONE ANALYSIS

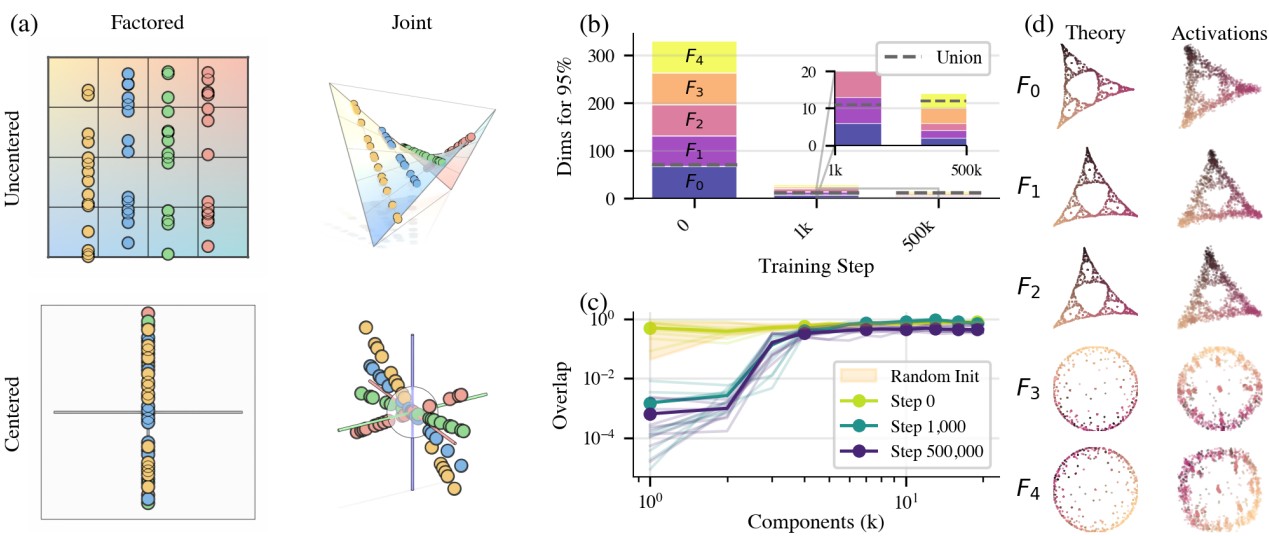

*Figure 15.* An expanded version of Figure 3 from the main text. a) A cartoon depiction of how we generate the vary-one dataset and how we use mean centering to isolate the variation due to each factor on its own. d) **The 2d subspaces we find through the vary-one analysis are enough to capture most of the predictive vector geometry.** Regression to the ground truth predictive vectors, using activations that come from driving the transformer on generated according to the true process and then filtering them through the first 2 PC's discovered by the vary-one analysis. Panels b) and c) are the same as in the main text.

For generative processes with $N$ independent factors, we can generate data where only one factor's subsequences vary while the others are held fixed. This enables us to probe the dimensionality of the subspace spanned by activation variance attributable to a single factor, and to extract orthonormal bases for use in the orthogonality analyses of Section H.2.

**Procedure.** For each factor $n$, we construct datasets where only factor $n$'s subsequences vary, keeping the subsequences of the other $N - 1$ factors fixed. For each fixed configuration of the non-varied factors, we collect and mean-center the activations from an ensemble of realizations of the varied factor. We then aggregate across many fixed configurations and perform PCA. The top $k$ principal components (chosen, e.g., to capture 95% of variance) define an orthonormal basis for a $k$-dimensional candidate subspace for factor $n$. Repeating this procedure for each factor yields $N$ candidate subspaces with orthonormal bases suitable for computing the overlap metric of Section H.2.2.

**Why mean-centering works.** Because the factors are independent, activations within each fixed configuration differ only due to variation in factor $n$. Mean-centering removes the contribution of the fixed factors, isolating the geometry associated with factor $n$.

**Checking factoredness.** The effective dimensionality of each factor's vary-one activations also provides a check on the degree of factoredness: for a fully factored representation, the effective dimensionality should match the intrinsic dimensionality of that factor's ground-truth belief simplex ($d_n - 1$). Higher effective dimensionality suggests the representation is not fully factored.

**Limitations.** Vary-one analysis requires the ability to generate samples by fixing the latent dynamics of all but one factor—something that is straightforward for independent factors but not for processes with conditional dependencies among factors. For such settings, we turn to regression-based subspace identification.

### H.1.2. REGRESSION-BASED SUBSPACE IDENTIFICATION

While vary-one analysis provides a direct method for identifying factor subspaces through controlled variation, it requires the ability to generate samples by fixing the latent dynamics of all but one factor in the generative process—something that

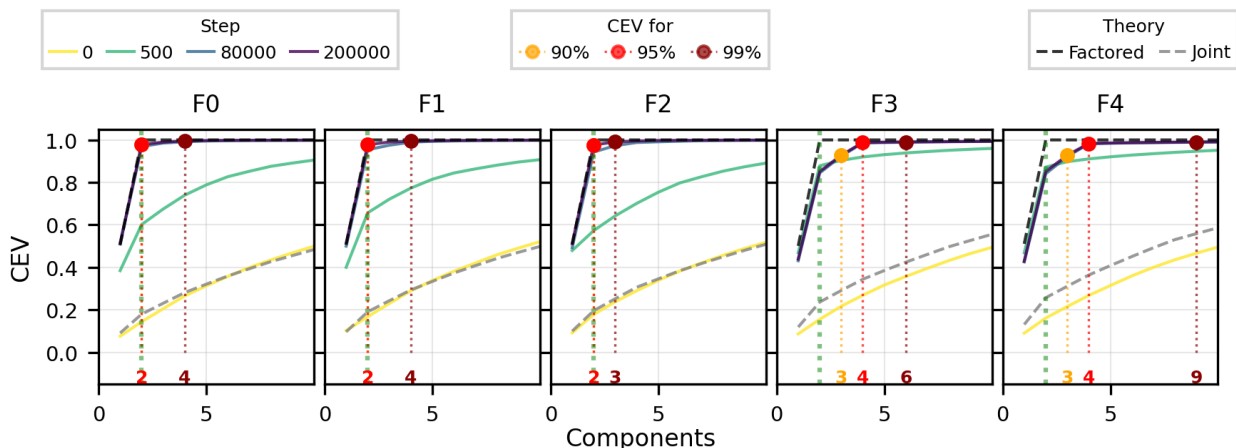

*Figure 16.* The CEV curves for residual stream activations when driven by the vary-one dataset for each factor. The vertical dashed green line represents the predicted dimensionality (2 for each factor, in this case). **As predicted, the first two components are responsible for much more variation than the rest for all factors**. Factors 3 and 4 ultimately spread out their activations across more dimensions, but still reach a CEV of 95% with only 4 PCs.

is straightforward for generative processes with independent factors but not for those with conditional dependencies among factors. In such settings, we can instead leverage the regression framework of Section G to identify factor subspaces.

As described in Section G.4, we perform a single joint regression from activations to the concatenated predictive vector $Y = [Y^{(1)}|\cdots|Y^{(N)}]$, yielding coefficient matrix $W = [W^{(1)}|\cdots|W^{(N)}]$ where $W^{(n)} \in \mathbb{R}^{d_{\text{model}} \times d_n}$ corresponds to factor $n$. We define the **factor subspace** $\mathcal{U}^{(n)}$ as the subspace spanned by the columns of $W^{(n)}$. To obtain an orthonormal basis for $\mathcal{U}^{(n)}$, we compute the SVD of $W^{(n)}$ and retain the left singular vectors corresponding to singular values above a numerical tolerance, yielding $Q^{(n)} \in \mathbb{R}^{d_{\text{model}} \times r_n}$ where $r_n \leq d_n$ is the effective rank. Orthonormalization is necessary because the overlap metric in Section H.2.2 relies on principal angles between subspaces, which are only well-defined with respect to orthonormal bases; without this step, the metric would conflate geometric alignment with the arbitrary scaling and skewness of the regression coefficients.

Intuitively, $\mathcal{U}^{(n)}$ captures the directions in activation space that carry information predictive of factor $n$'s predictive vector. If the transformer has learned a factored representation, we expect $\mathcal{U}^{(n)}$ to be approximately orthogonal to $\mathcal{U}^{(m)}$ for $n \neq m$. Note, however, that successful linear regression—low reconstruction error—does not by itself imply orthogonality; a representation in which factor subspaces overlap could still support accurate linear readout of each factor's predictive vector.

## H.2. Orthogonality analysis

### H.2.1. EFFECTIVE DIMENSIONALITY COMPARISON

We offer a global perspective on subspace orthogonality that leverages the relationship between the dimensionality of factor-specific representations and the dimensionality of their combined representation. For subspaces $\mathcal{A}$ and $\mathcal{B}$ of a common vector space, a standard result from linear algebra gives:

$$\dim(\mathcal{A} + \mathcal{B}) = \dim(\mathcal{A}) + \dim(\mathcal{B}) - \dim(\mathcal{A} \cap \mathcal{B}) \tag{42}$$

where $\mathcal{A} + \mathcal{B}$ denotes the subspace spanned by the union of $\mathcal{A}$ and $\mathcal{B}$. Thus, dimensionalities of the subspaces are additive only when $\mathcal{A} \cap \mathcal{B} = \{0\}$, i.e., when the subspaces are orthogonal. Rearranging Eq. (42), we see that $\dim(\mathcal{A}) + \dim(\mathcal{B}) - \dim(\mathcal{A} + \mathcal{B}) = \dim(\mathcal{A} \cap \mathcal{B})$, so any deviation from additivity directly measures the dimension of overlap of these subspaces.

In practice, activations form point clouds that approximately span low-dimensional subspaces rather than lying exactly on them. We therefore use the effective dimensionality $k_p^*$ as an empirical proxy for subspace dimension. For a transformer trained on a generative process with $N$ factors, we compute:

1. $k_p^{*(n)}$: the effective dimensionality of activations generated from vary-one datasets for factor $n$, mean-centered per frozen context as described in Section H.1.1

2. $k_p^{*(\text{all})}$: the effective dimensionality of activations from the union of all vary-one datasets.

If the factor subspaces are orthogonal, the sum $\sum_{n=1}^{N} k_p^{*(n)}$ should equal $k_p^{*(\text{all})}$. If the sum exceeds $k_p^{*(\text{all})}$, the subspaces must share directions. The magnitude of this discrepancy provides a global measure of overlap that complements the pairwise subspace overlap metric of Section H.2.2. While the overlap metric examines geometric relationships between explicitly identified bases, the dimensionality comparison requires only the effective dimensionality of activation point clouds, avoiding the need to commit to specific basis vectors.

The choice of CEV threshold $p$ introduces some arbitrariness into this analysis. In practice, we report results using $p = 0.95$ as we find that qualitative conclusions are robust across reasonable threshold choices (e.g., $p \in [0.90, 0.99]$) and that $p = 0.95$ more than captures the bulk of the variance associated with a given factor's predictive geometry.

### H.2.2. SUBSPACE OVERLAP METRIC

We provide a second, pairwise perspective by quantifying the orthogonality between two candidate subspaces $\mathcal{A}$ and $\mathcal{B}$ via their principal angles (also known as canonical angles). Given orthonormal bases $Q_\mathcal{A}$ and $Q_\mathcal{B}$ for each subspace (obtained via the procedures in Section H.1), we compute the singular value decomposition of their interaction matrix $M = Q_\mathcal{A}^\top Q_\mathcal{B}$. The singular values $\sigma_i$ of $M$ correspond to the cosines of the *principal angles* $\theta_i$ between $\mathcal{A}$ and $\mathcal{B}$. We define the **subspace overlap** as:

$$\text{overlap}(\mathcal{A}, \mathcal{B}) = \frac{1}{d_{\min}} \sum_{i=1}^{d_{\min}} \sigma_i^2 = \frac{1}{d_{\min}} \|M\|_F^2 = \frac{1}{d_{\min}} \text{Tr}(P_\mathcal{A} P_\mathcal{B}) \tag{43}$$

where $d_{\min} = \min(d_\mathcal{A}, d_\mathcal{B})$ is the dimension of the smaller subspace and $P_\mathcal{Y} = Q_\mathcal{Y} Q_\mathcal{Y}^\top$ is the orthogonal projection onto subspace $\mathcal{Y}$. This yields an overlap score between 0 and 1, which is a basis-invariant measure of the shared dimensionality between the subspaces, where 1 indicates that the smaller subspace is fully contained within the larger one, and 0 indicates that the subspaces are geometrically orthogonal. As suggested in Eq. (43), there are many ways to interpret this metric and it is mathematically equivalent to 1) the squared Frobenius norm of $M$ scaled by $\frac{1}{d_{\min}}$ and 2) the normalized trace of the product of the projection matrices, $\frac{1}{d_{\min}} \text{Tr}(P_\mathcal{A} P_\mathcal{B})$.

## I. Empirical synopsis

We find that neural networks do indeed learn to factor processes into parts, with latent distributions over these parts updated in parallel. We found (from CEV analysis) that only the low number of dimensions are used by these trained models. And we found that a linear map exists from these activations to the anticipated belief geometries in orthogonal subspaces, as suggested by the RMSE (reducing over the course of training together with loss) between the ground-truth geometry of the factors and the linear regression of activations to this geometry. Even without linear regression, the anticipated geometry can be seen in the relevant 2-dimensional subspaces defined by pairs of principal vectors.

