# OpenReview forum: "Transformers learn factored representations"
_ICML.cc/2026/Conference — ICML 2026 regular_

### Official Review · Reviewer_8ifN · 2026-03-06

**Soundness:** 4
**Presentation:** 3
**Significance:** 3
**Originality:** 3
**Overall Recommendation:** 5
**Confidence:** 3

**Summary:**

The authors train transformers with normal next token prediction on data generated by a factorable process, and find models also learn factorized representation and still tend to learn it even when doing so would cause loss of fidelity. More specifically, the data generation process is having multiple independent HMMs running, their outputs are then combined as tokens. Importantly, from the model’s side, there’s no explicit signal of underlying factorable structure, as it receives the combined outputs as a single token at each time step. The paper shows experiments where the dependency of factors varies, the model size and architecture varies etc, and shows the resulting geometry inside the model. These experiments and analysis provide clear evidence about models tend to learn factored representations.

**Compliance With Llm Reviewing Policy:**

Affirmed.

**Key Questions For Authors:**

1. What’s RMSE in line 271 on the right column? Can't find where it is introduced.
2. The paper shows the inductive bias is not specific to transformer architecture, do you think this is true for neural networks in general?
3. What’s your thoughts on [1]? It seems highly related. The paper emphasizes orthogonal subspaces in transformers as well but in very different setting. Do you think findings in that paper and yours somehow support each other?

[1]: Huang, X., & Hahn, M. (2025). Decomposing Representation Space into Interpretable Subspaces with Unsupervised Learning. arXiv preprint arXiv:2508.01916.

**Limitations:**

yes

**Strengths And Weaknesses:**

### Strengths
* It brings an under-represented (if not novel) while important and useful inductive bias into people’s attention. Such preference for factored representation shows that transformers are more interpretable and internally organized than we expect. Treating representation space as multiple orthogonal subspaces, each of which encoding a separate part of the world, is very useful for interpretation and intervention, and the paper highlights why we should expect it to occur in the first place.

* Experiment are conducted in a very rigorous manner and the claims are well supported. The experimental setting allows fine-grained control and rigorous verification of the resulting geometry. The evidence clearly supports the conclusion. Though whether the same finding would generalize to transformers trained on real-world data is another question.

* Overall, the presentation is clear. Though there are things can be improved (See suggestions)

### Weaknesses:
* The scale and diversity of the data generation process is limited. Throughout the paper, factors are only 5 GHMMs. I understand it’s hard to come up with a very different or a more real-world-related experiment setting while keeping the same rigor and analysis method, but having such settings would help a lot on making the paper more impactful. For example, can you try come up with other kinds of generative models and still find similar pattern? Or find certain kind of real world data modeled very well by independent HMM running in parallel?

### Suggestions:
* If accepted, authors can use the extra space to talk a bit more about (Shai et al, 2024) in early part of the paper, such as expanding the content around line 123. I’ve read that paper and am recalling its content to understand this one, but for general audience they might haven’t read it and need more introduction of it.
* Also, authors can give a more detailed description on the difference between the data generation process of 4.1 and that of 4.2

---

> ### Author Rebuttal · Authors · 2026-03-31
>
> We thank the reviewer for their thorough assessment, and are glad the rigor of the experimental design and the clarity of the evidence came through. We are particularly pleased that the reviewer found the framing around factored representations and their implications for interpretability compelling. We also appreciate the suggestion to expand the background on Shai et al. 2024 earlier in the paper; we agree this would help orient readers less familiar with that work, and it would indeed be a natural use of the extra space if accepted. Before diving into the question responses, we’d like to make a comment about connecting our toy model work to real-world scenarios.
>
> Connecting these findings to real-world data is something we are actively pursuing as a next step. As the reviewer points out, the focus of the present work is deliberately narrow: by using synthetic data with known ground-truth structure, we can make falsifiable predictions and verify them rigorously. Relaxing this control in favor of more naturalistic data would make the geometric analysis substantially harder to interpret. That said, we want to emphasize that GHMMs are an exceptionally broad class of generative models as essentially any discrete stochastic process with finite statistical complexity can be represented as a finite-state GHMM. A product of conditionally independent GHMMs running in parallel is therefore not as artificial a setting as it may appear. The practical challenge is not whether such decompositions exist in real data, but identifying the appropriate model parameters/dimensionality for a given dataset — a highly nontrivial problem that we regard as important future work. Demonstrating these patterns in a real-world setting will significantly increase the impact of the work, and we see our theoretical framework as laying the groundwork for exactly that investigation.
>
> ## Question Responses:
>
> 1. RMSE is the root mean squared error. In this case, this is the RMSE between two sets of vectors – namely, estimated predictive vectors $\hat{\eta}$ (found from regressing activations to ground truth predictive vectors) and ground truth predictive vectors $\eta$. We compute this as $\sqrt{\frac{1}{Nd}\sum_{n=1}^N\sum_{i=1}^d(\eta_i^{(n)} - \hat{\eta_i}^{(n)})^2}$.
> 2. We feel there is still more to understand about why this inductive bias seems to exist for transformers. For neural networks in general, this may depend largely on training tasks and class of model. For instance, with bidirectional attention and masked language modelling, the token prediction can be conditioned on both the future and the past. In this regime – because the model still needs to maintain beliefs over latent generative factors – we still expect factoring to occur, but the geometry will be different. For discriminative models in general, the model only needs to track enough features to infer a decision boundary, and the tracking of latent generative factors may be much weaker. We think this exploration is an exciting avenue for future research!
> 3. Thank you for bringing our attention to the recent preprint! While it’s not clear how comparable the synthetic settings are, their findings in the LLM setting seem to show that conceptually similar activations are organized into their own orthogonal subspaces. Our factored story provides a hypothesis for why this occurs – that transformers have an inductive bias towards belief updating in orthogonal subspaces when a process admits a (approximately) conditional factorization. So, if LLMs track beliefs over certain partitions of the world (e.g. concepts) separately and still do well on the loss, it is natural for them to do this in orthogonal subspaces. The predictions that our framework provides meant that it was a relatively simple matter to probe for them in the trained models. In general, some kind of unsupervised tool will be necessary to discover these subspaces; in this light, their findings also provide confidence that we may be able to leverage insights from our factored results to design even more effective tools for discovering interpretable subspaces in more real-world settings.

---

> > ### Author Rebuttal · Reviewer_8ifN · 2026-04-01
> >
> > I thank the authors for their response and for taking the time to address the comments. Their clarification helps improve the understanding of the work.

---

> > > ### Author Response · Authors · 2026-04-06
> > >
> > > We thank the reviewer for their strong recommendation, and appreciate their ideas for how to use the additional space in the revision.

---

### Official Review · Reviewer_KggH · 2026-03-11

**Soundness:** 3
**Presentation:** 4
**Significance:** 2
**Originality:** 2
**Overall Recommendation:** 4
**Confidence:** 3

**Summary:**

The paper studies the representations of transformers when applied to structured sequence learning tasks. More concretely they consider token sequences generated from a latent process which can be decomposed into independent sub-processes. These can be represented either by the independent dynamics (which is low dimensional) or as a complex dynamics on the product space (which is generally of higher dimension). While it is known that transformers can learn complex conditional probability distributions, it is not clear which representation they use. The experiments in the paper show that the more compressed representation can be found in the trained transformer and this remains true when the dynamics is slightly perturbed so that factorized representation can only approximate the full dynamics.

**Compliance With Llm Reviewing Policy:**

Affirmed.

**Final Justification:**

I believe this work makes a novel contribution to understanding Transformers. While it's scope is somewhat limited, I lean towards recommending acceptance.

**Key Questions For Authors:**

* Why are the sequence lengths 8, 33, and 101 considered?

* In Figure 3 (b): Why is the metric close to one for k=3? The corresponding spaces should still contain the orthogonal 2 dimensional spaces?

* Is the considered factorized dynamics novel or has it been studied before?

* Why are latent dynamics with a fractal dynamics considered? Would this work for simpler models?

**Limitations:**

The limitations of the results (generality, robustness) could be discussed in more detail.

**Strengths And Weaknesses:**

**Presentation**: The paper is very clearly written, well-structured, and has many helpful figures. However, there are no references in the introduction and the relation to prior work is only discussed in the related work section.

**Significance**: Understanding the structure of representations in transformer models is an important problem. The factorisation into independent mechanisms is a relevant assumption in this context. The results provide intuition about the structure of representations (dimensionality, orthogonality,…) in this case, which might be helpful for people working, e.g., on interpretability of transformer models. In addition, the observation that the projection on the factored dynamics is learned initially is interesting and relevant.

On the other hand, the results are not too surprising and can be seen as an instance of the very general observation that neural networks prefer simple representations. It is not clear whether the findings carry over to real data which might not have a similar fractal structure (or even different models from the same class). The paper does not provide an explanation for the observed representations and how this relates to architecture and optimisation. Overall, the findings of the paper are an interesting observation without proper explanation.

**Originality**: The data generating mechanism is interesting. Otherwise the approach seems to follow standard approaches for sequence learning (and extracting representations in transformer models).

**Soundness**: The theoretical presentation is very clear and convincing (especially for a mostly empirical paper). The metrics used seem to be generally appropriate. However, they focus on dimensionality and it is not always clear whether the subspace projections are sufficient to recover the dynamics of the corresponding component (this seems to be only shown for the full representation).
The experiments focus on one specific setting so that the generality of the claims remains a bit vague. Some numbers (e.g., sequence lengths) appear peculiar so the robustness of the claims is unclear.

---

> ### Author Rebuttal · Authors · 2026-03-31
>
> We thank the reviewer for their comments and appreciate the positive comments on presentation and clarity.
>
> **On lack of explanation:** We want to be clear about what we do and don't explain. We do explain why these representations arise. They are derived from first principles as the geometry that optimally tracks predictive information about each factor, and we predict them exactly before training begins. What remains unexplained is why gradient descent preferentially finds the factored solution over the joint alternative when both are available. We agree this is an important open question. The fact that RNNs and LSTMs show similar behavior (Sec. 4.4) suggests the answer may involve generic inductive biases of gradient descent rather than architecture-specific mechanisms; a direction we are actively pursuing.
>
> **On originality:** while our task setting is standard next-token prediction, a number of aspects of our work are not standard. Most importantly, we introduce a theory to specify, before training, two hypotheses for the exact features that should emerge, and the precise geometric relationships amongst them. Our analysis makes use of linear probing, dimensionality estimation, and the vary-one process. The latter is only possible because of our ground-truth generator, which the reviewer identified as a strength.
>
> **On related work:** We agree that our work relates to previous work about networks forming compact representations. However, we provide a quantitative, and falsifiable theory specifying the exact geometric structure of activations, in particular focusing on the direct sum factorization between multiple multi-dimensional subspaces. Additionally, the inductive bias result in S4.3 where models prefer factored representations at measurable cost to predictive fidelity, goes beyond the general simplicity picture. A model that merely preferred simplicity would compress into fewer dimensions. What we show is that the model prefers a *specific* geometric structure, the factored direct sum, over other specific geometries that could offer better prediction. The model is not just compressing; it is factoring. We will add discussion of Trager et al. (2023), whose Corollary 8 establishes a closely related connection, and clarify how our framework extends it. We also agree the submission would benefit from more serious placement in the context of prior work on factorization and disentanglement; see our responses to reviewer Ebt7 for details.
>
> **On subspace dynamics:** the projections do recover individual component dynamics. Linear regression from the 2D vary-one subspace to each factor's ground-truth predictive vectors achieves high fidelity (Fig. 14d, Appendix H.1.1). We will make this explicit in the main text.
>
> **Question Responses:**
>
> 1. The values 33 and 101 carry no special significance. They are simply 32 and 100 plus a BOS token. Sequence length determines how much belief space is explored during training. By length 9 (8+BOS), appreciable latent structure comes into play. Longer contexts verify the solution does not qualitatively change.
> 2. Note the log scale. At k=2, overlap is below 0.003; at k=3 it jumps to \~0.3. This is consistent with each factor using exactly 2 orthogonal directions: the third principal component from one factor aligns with directions from other factors, producing the expected jump. This confirms the predicted 2D subspace structure.
> 3. The general finding that networks learn structured representations is not new, and we do not claim otherwise. However, "factored" is used loosely across the literature. Hinton's distributed representations say nothing about orthogonality or dimensionality. The disentanglement literature typically assumes 1D factors or introduces explicit bottlenecks/KL penalties; we apply no such engineering. Recent geometric embedding work focuses on dot-product relational structure rather than direct sum factorization. Our contribution is a precise mathematical definition of factorization in the sequential predictive setting, a theory deriving the expected geometry from the data-generating process, and the finding that optimization selects this geometry even when lossy.
> 4. Fractal geometry arises as the generic outcome of nonunifilar generators, where knowing the current latent state and the emitted token does not determine the next latent state. This forces the observer to maintain distributions over states that spread out over the predictive space producing the continuous belief manifolds with fractal structure that our experiments test. Nonunifilarity is the rule rather than the exception for structured sequential data, likely including natural language. In contrast, unifilar generators, where state plus observation determines the next state, lead to much more point-like predictive geometries, and thus are not suitable tests for the precise non-orthogonality of embeddings that track similarity in future predictions between different contexts.

---

> > ### Author Rebuttal · Reviewer_KggH · 2026-04-03
> >
> > I appreciate the authors addressing my comments and concerns.
> >
> > Based on the authors feedback, I am now leaning towards acceptance and will update my rating accordingly.

---

> > > ### Author Response · Authors · 2026-04-06
> > >
> > > We are glad that the reviewer found our clarifications helpful. We also believe that adding a few points of emphasis, inspired by the review, in the revision will improve the quality of our presentation.
> > >
> > > We do not see the overall review updated to recommend acceptance in the OpenReview portal, and just wanted to clarify whether the reviewer was still intending to do so.

---

### Official Review · Reviewer_dfES · 2026-03-12

**Soundness:** 3
**Presentation:** 4
**Significance:** 4
**Originality:** 4
**Overall Recommendation:** 4
**Confidence:** 4

**Summary:**

This paper investigates whether Transformers, trained via standard next-token prediction, inherently learn to decompose complex sequence data into independent latent factors. The authors formalize two hypotheses for internal activation geometry: a "joint representation" whose dimensionality scales exponentially with the number of parts, and a "factored representation" that scales linearly by placing factors into orthogonal subspaces. By constructing synthetic datasets using generalized Hidden Markov Models (GHMMs) with known ground-truth geometries (e.g., Mess3 and Bloch Walk processes), the authors empirically verify that Transformers naturally converge to factored representations when factors are conditionally independent. Furthermore, the paper proposes that Transformers possess a fundamental inductive bias for factorization, demonstrating that models will adopt a dimensionally efficient but lossy factored representation early in training even when injected noise breaks the conditional independence of the underlying data.

**Compliance With Llm Reviewing Policy:**

Affirmed.

**Final Justification:**

This paper presents a unique observational perspective on the interpretability of transformer features, and the experimental process is rigorous. However, I am concerned about the generalizability of the experiments in scenarios closer to real-world complexity.

**Key Questions For Authors:**

See the weaknesses

**Limitations:**

yes

**Strengths And Weaknesses:**

Strengths:
1) Rather than relying on post-hoc feature extraction, the authors provide a rigorous theoretical framework that analytically predicts the exact multidimensional geometric structures (e.g., simplices and Bloch spheres) of model activations based on the underlying data-generating process..
2) The mathematical formulation bridging GHMMs, predictive vectors, and product-state manifolds is clearly articulated.
3) The paper is well-written and structured.

Weaknesses:
1). Real-world data distributions are typically characterized by high complexity and dimensionality. The current experimental paradigm assumes that the residual stream's dimension is sufficiently large to comfortably accommodate the orthogonal embeddings of the entire vocabulary. However, it remains unclear how the model's behavior changes when the theoretical dimension of the joint representation (or the number of factors) strictly exceeds the model's physical capacity, thereby creating a "representation bottleneck." Does the optimization trajectory for discovering factored subspaces change under such capacity constraints?
2). In Section 4.3, to demonstrate the inductive bias that the model "prefers factorization even at the expense of predictive fidelity," the authors introduce pure random uniform noise (token replacement). Considering pure noise inherently represents a complete loss of information that is unlearnable even by an optimal joint representation, is it reasonable to introduce pure random uniform noise?

---

> ### Author Rebuttal · Authors · 2026-03-31
>
> We thank the reviewer for their thoughtful engagement and constructive feedback, and are glad our mathematical framework that uses ground-truth generative structure to make concrete, testable predictions about activation geometry came through clearly. Both weaknesses raise good opportunities to clarify aspects of the paper that are already addressed in our experiments but not emphasized or explained clearly enough in the main text. We respond to each below and will incorporate these clarifications into the revised manuscript.
>
> 1) Regarding the regime of d\_model \< d\_joint: In fact, we do test this regime. The full joint representation for our primary test process is 243 (3^5-1). Our primary highlighted results come from a model with d\_model=120, which is strictly less than that. We also tested models from a wider sweep of parameters. In our sweep, we include models (transformers and RNN’s) from d\_model=32 to d\_model=480, which is large enough, even, to one-hot encode the entire token alphabet. In all cases, we find that our predictions hold. With this in mind, our results do show that, whether the residual stream is larger or smaller than the dimension of the joint representation, the transformers learn the factored representation. That being said, this is not quite small enough to probe the breakdown of the factored representation itself. This limit is definitely something worth exploring in the future, but our present work is specifically targeted on the question of joint versus factored.
> 2) Regarding the learnability of the noised process: this is a very important point to clarify so thank you for bringing it up. In our experiments we introduce noise such that, with probability epsilon, each token is replaced by a token uniformly sampled from the alphabet. Therefore, we can think of there being two regimes, one where 0 \< epsilon \< 1 and one where epsilon \= 1\. At epsilon \= 1 the process becomes, as you say, pure noise and the minimal generator becomes a one state HMM that uniformly emits over the joint vocabulary. In our experiments we only consider the first regime where 0 \< epsilon \< 1\. The key subtlety is that the observer does not know which tokens were corrupted. After observing a token, the posterior belief is a mixture. With weight (1−epsilon) the token was faithful and each factor updates independently. With weight epsilon it was noise and uninformative. Because this uncertainty applies jointly across all factors, it entangles beliefs about different factors, pushing predictive vectors off the product-state manifold in a way the factored representation cannot capture. Increasing epsilon strengthens these cross-factor correlations, creating increasing pressure on the transformer to represent joint beliefs faithfully. This is reflected empirically in the increasingly higher dimensional representations we can see in the evolution of the CEV curves over training (Figure 12\) as we increase epsilon. Notably, despite the true predictive vectors living off the product state manifold, we find that for all values of epsilon, the transformer first learns representations consistent with the (sub-optimal) low dimensional factored beliefs. Thus, the noise regime we study is precisely the regime where the inductive bias claim is nontrivial: there *is* structure to learn, but the model chooses a factored approximation over a more faithful joint representation. It is this choice that we claim suggests an inductive bias towards factored representations over predictive fidelity.

---

> > ### Author Rebuttal · Reviewer_dfES · 2026-04-04
> >
> > Thank you for the response. The author explanations clearly elucidate the experimental setup, particularly regarding the noise injection mechanism (clarifying that epsilon was used to create a mixed posterior rather than a pure noise scenario). It is now clear to me that this design is quite reasonable and effectively entangles the factors without making the sequence completely unlearnable. Regarding the capacity constraints, I appreciate the clarification that the parameter sweeps included models where d_model is strictly less than d_joint. However, as the authors themselves noted in the rebuttal, this scale may still not be small enough to fully probe the breakdown limit of the factored representation itself. One could imagine a more rigorous exploration of extreme capacity bottlenecks to see exactly where and how the inductive bias for factorization truly collapses. In any case, the clarifications are well-received and definitely address my primary confusions. In light of this, I would like to keep my original score.

---

> > > ### Author Response · Authors · 2026-04-06
> > >
> > > We thank the reviewer for the positive feedback and the high subcategory scores, especially with regard to significance, originality, and presentation.
> > >
> > > We also appreciate the reviewers idea to further investigate what happens when even the factored solution is too large for the model. The cases in which the factored representation fails is an important piece of follow-up work that the current framework facilitates. In the revision we will make sure to emphasize that our claims pertain specifically to cases where the model has capacity for the factored representation.

---

### Official Review · Reviewer_Ebt7 · 2026-03-12

**Soundness:** 3
**Presentation:** 3
**Significance:** 2
**Originality:** 2
**Overall Recommendation:** 4
**Confidence:** 3

**Summary:**

This paper contrasts two extreme types of representation: one based
on the product space and one based on an orthogonal factorization.
When factors are conditionally independent, the factorization is
efficient. Even when models have capacity to learn a product-space
representation, this paper argues that transformers have an inductive
bias favoring the factored representations. Further, when conditional
independence is violated, models still learn factored representations,
reflecting the strong inductive bias. Empirical results are presented
demonstrating the evolution of factorized representations during training.

**Compliance With Llm Reviewing Policy:**

Affirmed.

**Final Justification:**

Author rebuttal was helpful in appreciating the relation to related work. I am still unclear how the present work informs the basic story already in the literature that neural nets generally have an inductive bias to learn factored representations, and I don't see how this work gives us the why or any further guidance. Still, the authors have been relentless in their responses, and I am hoping others will get more from the work than I did, so I raised my score.

**Key Questions For Authors:**

My main concern is that the result presented here seems too similar to other literature on learning geometric embeddings. This paper may be more or less contemporaneous but seems at its core related and also makes the point that even when compression isn't required, geometric embeddings are learned:  https://arxiv.org/abs/2510.26745.  But even going back to the dawn of back propagation, the first papers all emphasized learning of distributed representations. For example, the Hinton family trees paper (https://www.cs.toronto.edu/~hinton/absps/families.pdf) circa 1987 showed discovery of factorized representations instead of product-space representations (although in that work there was a bottleneck encouraging factorized representations). Also, there's the literature on compositionality and disentangling of representations that was super popular in the vision literature circa 2018. Could the authors help me to understand what this work adds to the mix?


"Given the theoretical underpinnings of our empirical results,it is plausible that the factored world hypothesis extrapolates to
larger models as well." Don't the well known results on linear steering vectors already tell us that representations are being factored? It's hard to imagine transformers working at all otherwise.

You write, "our theoretical framework makes precise predictions about the geometry of transformer activations..." (line 260-264) Are you making this claim qualitatively or does your framework allow for quantitative predictions?  In all result figures, I see empirical results but not matched theoretical predictions. A qualitative theory is better than no theory, but I was really hoping to see quantitative predictions, e.g., how degree of conditional dependence affected factorized representations.

**Limitations:**

Yes

**Strengths And Weaknesses:**

Strengths:
* The visualizations are pretty cool.
* The figures do a good job explaining the results.
* The article is well written, although I had a bit of trouble parsing the abstract (the critical sentence beginning "We formalize..."). Also, the paper emphasizes next-token prediction as the source of factorization, but it seems a more encompassing finding since it exists in the literature in many domains and for many training objectives (e.g., word2vec, supervised learning).

Weaknesses:
* I did not gain a lot of insight from the paper. The bias toward factorization seems evident in the literature for a wide variety of neural net architectures already.
* The paper seems to oversell the consequences of the theory, but possibly I'm not understanding (see questions below).

---

> ### Author Rebuttal · Authors · 2026-03-30
>
> We agree that the existence of structured representations in neural networks is not a new finding. Our contribution is a theory that allows us to derive competing geometries associated with the same dataset, each with distinct predictions for dimensionality, subspace organization, and internal structure. This lets us not just ask whether the model factors, but anticipate what the factors should be, and whether the model prefers factoring over the joint alternative. Crucially, our theory predicts exactly when features of the data make factoring suboptimal relative to tracking the joint. In this regime, we find transformers collapse to the factored solution early in training and dwell there, sacrificing measurable predictive accuracy, and expand towards the optimal joint representation only under sustained gradient pressure. Without our theoretical framework, it would be difficult to distinguish factoring-as-optimality from factoring-as-inductive-bias.
> ## Prior Work
> - The reviewer is right that there is a structural parallel to Hinton (1986), where product-space representations are exponential in the number of parts and factored ones are linear. However, in that work the product-space representation is a one-hot lookup table and as the reviewer notes, the bottleneck makes it infeasible regardless. In our setting, the joint representation is Bayesian optimal for prediction with rich internal geometric structure. The finding that transformers do not learn it, and prefer the factored alternative even at a cost of higher loss, is novel. The sequential predictive setting also introduces dynamics: product states must remain product states under belief updating as context grows. This is what generates the specific dimensionality and geometric predictions our experiments test.
> - Trager et al. (2023) is the closest prior work and should be cited. Their Corollary 8 establishes the same core connection: embeddings are linearly factored if and only if factors are conditionally independent. We extend their result into the autoregressive setting, derive a competing joint geometry, and test what happens when conditional independence is violated (Sec. 4.3).
> - Noroozizadeh et al. (2025) shares our finding that geometric structure emerges without capacity pressure. However, their geometry is entity embeddings reflecting graph topology (Fiedler vectors of the graph Laplacian), quite different from direct-sum factorization of predictive states.
> - Locatello et al. (2019, ICML Best Paper) proved unsupervised disentanglement is impossible without inductive biases; our work identifies a setting where the task structure itself provides the bias. The disentanglement literature (β-VAE, etc.) studies how to engineer factored representations through modified objectives; we study what standard training selects with no such incentive, and our factors are multidimensional subspaces with internal structure rather than 1-dimensional latents.
> ## Steering
> We agree that effective steering with fixed directions is suggestive of factored structure. But steering operates on individual directions and doesn't tell you the dimensionality of each factor's subspace, the internal geometry within each subspace, whether subspaces are orthogonal to each other, or whether the model would maintain this structure even when it hurts prediction. Our framework specifies these, and our experiments test them. Steering vectors are consistent with our findings but underdetermined, as many geometries could support effective steering.
> ## Quantitative Predictions
> Matched theoretical predictions are present in the figures, and we should have labeled them more clearly. In the regime of conditionally independent factors, our theory makes exact quantitative predictions, including the total effective dimensionality, the number of orthogonal subspaces, the dimensionality within each subspace, and the internal geometry within each subspace given by the predictive states of that factor. These predictions are already present in the figures: the dashed curves in Figure 2 and 9 are the theoretical baselines for factored (10D) and joint (242D) representations, and the empirical curves converge to match the factored prediction. The within-subspace geometries in Fig. 2a (left panels) are also theoretical, matched against empirical activations (right panels) with RMSE quantified in Fig. 8. We should have labeled this comparison more prominently and will improve the presentation.
>
> In the noisy regime, our theory correctly predicts the geometry transformers converge to and dwell at early in training, but does not predict the exact geometry they ultimately reach after expanding under gradient pressure. A theory capturing this tradeoff is a main direction of future work.
>
> We appreciate the reviewer's directness in pushing us to position this work more carefully relative to the prior literature. We will sharpen the scoping of our claims and expand the related work discussion in revision.

---

> > ### Author Rebuttal · Reviewer_Ebt7 · 2026-04-03
> >
> > Thanks to the authors for their thorough and coherent response to my questions.
> >
> > When the authors state that they have shown quantitative predictions in their figures, I need help understanding what they mean. I appreciate that, for example, Figure 2a shows the generative factors in the first column and the model training in thge second column. I don't consider this to be a comparison of theory vs. actual behavior, though. What I'm looking for is a case where the theory informs exactly where and when factorization takes place, or how much the data must deviate from a factored representation before the model fails to learn a factored representation.
> >
> > I'll bump my score up, as I am convinced that the work is somewhat distinct from other work (and the most relevant other work is probably contemporaneous).

---

> > > ### Author Response · Authors · 2026-04-06
> > >
> > > We are glad the rebuttal was helpful and appreciate the reviewer's willingness to update their score. We'd like to clarify a few points in response to the remaining questions. We do quantitatively predict when models factor (when transition operators are conditionally independent) and also how models factor: the number of subspaces, the dimensions of each, and the nontrivial geometry of how context-induced activations are arranged within each subspace. We then compare those theoretical predictions to actual trained models, and find agreement. The two columns that the reviewer refers to represent nontrivial predictions in their own right. The left column is not simply "showing the generative factors" but a particular geometry of the predictive vectors induced by the process that we expect to find in the residual stream. If we intended simply to show the generative factors, we would have shown schematic-level generative finite state machines.
> > >
> > > The reviewer is asking for an additional quantitative prediction: when the factored approximation breaks down. Empirically, Figure 12 already shows this transition: when parts are coupled, the model learns the predicted factored representation before expanding into higher-dimensional representations under gradient pressure. We currently do not have a closed-form prediction of the boundary, but we can use the relative entropy between next-token distributions induced by the factored vs the joint representation to anticipate the loss gap at which the factored approximation becomes untenable. We will include this prediction in an appendix of our revision. A full theory of how this expansion takes place, and which approximations are shed first, second, etc., is a theoretically rich and substantial project that the current framework makes possible, and one we are actively pursuing.

---

### Decision · Program_Chairs · 2026-04-30

**Decision:**

Accept (regular)

**Comment:**

The review committee reached a positive consensus to accept the paper.
The reviewers praised the work for its clear presentation, rigorous experimental design, and precise quantitative predictions regarding activation geometry.
The framework provides valuable insights for the field of mechanistic interpretability.

While reviewers noted the limitations of using synthetic toy data rather than real-world datasets, they agreed that the mathematical work  here is a significant contribution to the field.
During the rebuttal phase, the authors successfully addressed concerns regarding model capacity bottlenecks, the specific mechanics of the noise injection regime, and the work's distinction from previous literature on neural network compression.

As promised by the authors during the rebuttal and discussion phase, the final manuscript must include the following updates:
1. An addition to the appendix quantifying the loss-gap using the relative entropy between next-token distributions of factored vs. joint.
2. An expanded related work discussion to more carefully position the work, specifically citing Trager et al. (2023).
3. Expanded background context regarding Shai et al. (2024) earlier in the text.
4. Emphasizing that the paper's claims pertain specifically to cases where the model has the capacity for the factored representation.